


# Comparison and validation of global and regional ocean
# forecasting systems in the South China Sea
Xueming Zhu[1], Hui Wang[1], Guimei Liu[1*], Charly Régnier[2], Xiaodi Kuang[1], Dakui
Wang[1], Shihe Ren[1], Zhiyou Jing[3], Marie Drévillon[2]
[1]Key Laboratory of Research on Marine Hazards Forecasting, National Marine Environmental
Forecasting Center, Beijing, 100081, China
[2]Mercator Océan, Ramonville Saint Agne, France
[3]State Key Laboratory of Tropical Oceanography, South China Sea Institute of Oceanology, Chinese
Academy of Sciences, Guangzhou, 510301, China
*Correspondence to*: Guimei Liu (liugm@nmefc.gov.cn)
**Abstract.** In this paper, the performances of two operational ocean forecasting systems, Mercator Océan
(MO) in France and South China Sea Operational Forecasting System (SCSOFS) in China, have been
examined. Both systems can provide science-based nowcast/forecast products, such as temperature,
salinity, water level and ocean circulations. Based on the observed satellite and *in-situ* data have been
obtained in 2012 in the South China Sea, the comparison and validation of the ocean circulations, the
structures of the temperature and salinity, and some mesoscale activities are shown. Comparing with the
observation, the ocean circulations and SST of MO show better results than those of SCSOFS. However,
the structures of temperature and salinity of SCSOFS are better than those of MO. For the mesoscale
activities, SST fronts and SST decreasing during the typhoon Tembin of SCSOFS are better agreement
with the previous study or satellite data than those of MO; but both of them show some differences from
AVISO data. Finally, according to the results compared in above, some suggestions have been proposed
for both systems to improve their performances in the near further.
**Keywords.** SCSOFS, Mercator Océan, South China Sea, Operational Forecasting System
**1 Introduction**
The South China Sea (SCS, Fig.1) is the largest and deepest semi-enclosed marginal sea of the
Northwestern Pacific (NWP), with the area is about 3.5 million km$^2$, the mean and maximum depth is
about 1200m and 5300m, respectively. A wide continental shelf with depth less than 200m located in the
northern SCS (NSCS). There are numerous islands, reefs, beaches, shoals in large basin of the southern



SCS (SSCS). It is connected with the adjacent seas through a number of channels, to the East China Sea
in north, the NWP in east, the Sulu Sea in southeast, and the Java Sea in south, by the Taiwan Strait
(TWS), the Luzon Strait (LUS), the Mindoro Strait and the Balabac Strait, the Karimata Strait,
respectively. Its unique geographical features, rich marine mineral and petroleum resources play a
significant role to many countries adjacent to it.
The SCS is located in the East Asian Monsoon (EAM) winds regime, the northeastly winds usually
prevail with an average wind speed of 9m/s over the whole domain in winter, while the southwestly
winds prevail with an average magnitude of 6m/s dominating over the most parts of the SCS in summer
(Hellerman and Rosenstein, 1983). The EAM is considered to be the main factor for driving the upper
layer basin-scale circulation pattern in the entire SCS, showing an obvious seasonal variation with a
cyclonic gyre in winter and an anti-cyclonic gyre in summer (Wyrtki, 1961; Mao et al., 1999; Wu et al.,
1999; Qu, 2000; Chu and Li, 2000). However, some other literatures insist that a persistent cyclonic gyre
in the NSCS, while a biannually changing from cyclonic gyre in winter to anti-cyclonic gyre in summer
in the SSCS (Chao et al., 1996; Takano et al., 1998; Hu et al., 2000; Chern and Wang, 2003; Caruso et al.,
2006; Chern et al., 2010). Chern et al. (2010) suggested that the three dynamical processes, the wind
stress curl, the deep-water ventilation-induced vortex stretching in the central SCS, and a positive
vorticity generated from the left flank of the Kuroshio in the LUS, play the equal importance to the
formation of the persistent cyclonic gyre in the NSCS, according to the analysis of the results from
several numerical experiments with different wind stress, topography and coastline.
In addition to the basin-scale circulations, there are still some sub-basin scale currents in the SCS, such as
the Guangdong coastal current (Huang et al., 1992), the SCS Warm Current (SCSWC, Guan, 1978; Chao
et al., 1995), Dongsha Coastal Current (DCC, Su, 2005), Luzon Coastal Current (LCC, Hu et al., 2000),
and so on. However, there are still a lot of debates about the mechanisms of some of them among the
studies reported by several authors, without reaching an agreement. For example, based on the results of
the numerical simulations, the formation dynamical mechanism of the SCSWC may be related to the
Kuroshio intrusion (Li et al., 1993; Cai and Wang, 1997), sea surface slope (Fang and Zhao, 1988; Guan,
1993), or the wind relaxation (Chao et al., 1995).
The Kuroshio intrudes into the SCS through the LUS, carrying the warm and salty water from the NWP,
significantly affecting the circulation pattern and the budgets of heat and salt in the NSCS (Farris and
Wimbush, 1996; Wu and Chiang, 2007; Liang et al., 2008; Nan et al., 2013). However, it is still not in





accordance with how the Kuroshio intrudes into the NSCS. As pointed out in Hu et al. (2000), there
existed four viewpoints on the Kuroshio intrusion as, a direct branch from the Kuroshio (Williamson,
1970; Fang et al., 1996; Chern and Wang, 1998; Qu et al., 2000), a form of loop (Zhang et al., 1995; Liu
et al., 1996; Farris and Wimbush, 1996), a form of extension (Hu et al., 1999), and a form of ring (Li et al.,
1998a, b) at present. Nan et al. (2015) reviewed and summarized the Kuroshio intruding processes from
observed data, numerical experiments and theoretical analyses, and concluded that there were three
typical paths of the Kuroshio intruding the SCS, the looping path, the leaking path and the leaping path,
which could be distinguished quantitatively by a Kuroshio SCS Index (KSI, Nan et al., 2011a) derived
from the integral of geostrophic vorticity southwest of Taiwan. The three paths can change from one to
another in several weeks.
In addition, many mesoscale eddy activities are another obvious physical characteristics of the NSCS,
and play a great influence on the dynamical environment of the NSCS. Eddies are generally more
energetic than the surrounding currents and are an important component of dynamical oceanography at
all scales. In particular they transport heat, mass, momentum and biogeochemical properties from their
regions of formation to remote areas where they can then impact budgets of the tracers. Eddies in the
NSCS have attracted increasing attention over recent a few decades. Much work has been done based on
the combination of satellite observation and *in-situ* hydrographic data (Wang and Chern, 1987; Li et al.,
1998; Chu et al., 1999; Wang et al., 2003; Hu et al., 2011; Nan et al., 2011b), or numerical models (Wu
and Chiang, 2007; Xiu et al., 2010; Zhuang et al., 2010). Some of work has been focused on the statistical
characteristics of eddies in the SCS, but they are greatly different from each other, owing to different
criteria for eddy identification employed by different literatures (Wang et al., 2003; Xiu et al., 2010; Du
et al., 2014). Some of work analyzes eddies' seasonal variability (Wu and Chiang, 2007; Zhuang et al.,
2010) and investigates their genesis (Wang et al., 2008). Some of work mainly studies specific eddies to
better understand eddy's generation, development and disappearance mechanisms (Wang et al., 2008;
Zhang et al., 2013).
As shown above, the dynamic processes and relative mechanisms are very complex, but still not cleared
until now in the SCS. It will be much more difficult to predict the status of the future ocean. National
Marine Environmental Forecasting Center (NMEFC) is mainly responsible for the prediction of the sea
area of the South China Sea, has built a SCS Operational Forecasting System (SCSOFS). As is known to
all, the open boundary forcing plays an important role in the numerical prediction of the regional ocean.





Due to the various limitations, the current SCSOFS' open boundary conditions (OBC) are derived from
the Simple Ocean Data Assimilation (SODA) climatological monthly mean during the forecast run. It is
extremely inappropriate for the real-time ocean prediction system, so we plan to transform the OBC from
SODA to the real-time forecasting results derived from Mercator Océan (MO) on the next step, in order
to further improve prediction accuracy of the SCSOFS. Before carrying out this work, it is necessary to
compare and validate the performance of MO in the SCS.
The focusing of this paper will be the comparison and validation of the performances of MO and
SCSOFS in the SCS, based on the observation data we have got in 2012. The rest of this paper is
organized as follows. Section 2 gives the introductions to the observed data which are employed to
validate the systems, and the configurations of MO and SCSOFS. Section 3 shows the results of
comparison and validation and discussions. Section 4 presents the summary and conclusions.
**2 Observed data and numerical operational systems**
**2.1 Satellite data**
The Map of Sea Level Anomaly (MSLA) and Map of Absolute Dynamic Topography (MADT) data, also
with the relative Geostrophic Velocity Anomaly (GVA) and Absolute Geostrophic Velocity (AGV) data
derived from them, respectively, are used to analysis the mesoscale eddy in the SCS and compare with
the numerical simulations. They are all-sat-merged and gridded delayed-time altimeter product produced
by SSALTO/DUACS and distributed by Aviso in April 2014, with support from Centre National
D'études Spatiales (Cnes, www.aviso.altimetry.fr). The products are sampled on a $0.25°\times0.25°$
resolution Cartesian grid in both longitude and latitude from the Mercator gridded product, with a daily
temporal resolution. Its period covers from 1993 to present, and the period of reference has been changed
from 7 years (1993-1999) to 20 years (1993-2012). It has been corrected for instrumental errors,
environmental perturbations, the ocean sea state influence, the tide influence, atmospheric pressure and
multi-mission cross-calibration (CLS, 2015).
Two kinds of Sea Surface Temperature (SST) data will be used in this paper. One is derived from the
merged satellite (NOAA/AVHRR, MetOp/AVHRR, GCOMW1/AMSR2, Coriolis/WINDSAT) and
*in-situ* data Global Daily SST (MGDSST), with a $0.25°\times0.25°$ horizontal resolution, which is analyzed



and published by Japan Meteorological Agency (JMA). The data can be obtained from
http://near-goos1.jodc.go.jp/.
The other one is derived from the NOAA 1/4 ° daily Optimum Interpolation Sea Surface Temperature
(OISST), which is an analysis constructed by combining observations from different platforms, such as
satellites, ships, buoys, on a regular grid via optimum interpolation. Right now, National Centers for
Environmental Information (NCEI) provides two kinds of OISST: one uses infrared satellite data from
the Advanced Very High Resolution Radiometer (AVHRR) named as AVHRR-only, and the other one
uses AVHRR data along with microwave data from the Advanced Microwave Scanning Radiometer
(AMSR) on the Earth Observing System Aqua or AMSR-E satellite named as AVHRR+AMSR. The first
one, AVHRR-only, is used in this study, which spans 1981 to the present and can be downloaded from
the website http://www.ncdc.noaa.gov/oisst/data-access.
**2.2 *In-situ* data**
The *in-situ* data employed in this paper for the comparison and validation of both systems are provided
by the South China Sea Institute of Oceanology, Chinese Academy of Sciences. There were one mooring
to measure the sea water velocity and 5 cruises implemented to measure the temperature and salinity (TS)
in the SCS during 2012.
The mooring station is located at Maoming (Fig. 1), where bottom-mounted upward-looking 75 kHz
Acoustic Doppler Current Profilers (ADCPs) are deployed to monitor the current profile (U component
and V component) from the depth of 2m to 48m with a 2-m interval in vertical. The period of the
monitoring is from 11 July to 8 October, in 2012, with a temporal interval 10 min. Firstly, the abnormal
data are eliminated from the original measured data; in the second, a low-pass filter with 25-hour is
applied to filter out the tidal current; and a 25-hour average is calculated to get the daily average data, in
order to compare and validate with the simulated results of MO and SCSOFS.
The TS data from 5 cruises are measured by SeaBird 19 plus conductivity-temperature-depth (CTD) with
1-m resolution in vertical. Among the 5 cruises, one is the Qiongdong cruise in the NSCS, which was
conducted 9 days from 12 to 20 July at 90 stations along 6 sections (See Fig.1); another one is the Nansha
cruise around the Nansha Islands, which was conducted 5 days from 24 to 28 August at 17 stations along
10 °N section from 109.5 °E to 117.5 °E. The TS data from those two cruises will be used to compare and
validate the TS distribution from MO and SCSOFS in vertical and horizontal. All the measured TS data



from 5 cruises will be collected to compare with the simulated data from MO and SCSOFS via
correlation analysis.

**2.3 The configurations of SCSOFS**

The SCSOFS is build up based on the Regional Ocean Modelling System (ROMS), which is a
three-dimensional, non-linear primitive equations, free surface, hydrostatic, split-explicit,
topography-following-coordinate in vertical and orthogonal curvilinear in horizontal on a staggered
Arakawa C-grid oceanic model (Shchepetkin and McWilliams, 2005).
To avoid the influences of boundary to the circulations in the SCS, the model's boundaries was extended
to southward and eastward, then the model covered a larger domain (4.5 °S to 28.3 °N, 99 °E to 145 °E, Fig.
1) than the SCS. The horizontal resolution variates from 1/12 ° in the south and east boundary to 1/30 ° in
the SCS. There were 36 s-coordinate levels in the vertical with the thinnest layer being 0.16 m on the
surface. The bathymetry was extracted from the ETOPO1 data sets published by U.S. National
Geophysical Data Center (NGDC), which is a global relief model of Earth's surface that integrates ocean
bathymetry and land topography, with 1 arc-minute resolution (Amante and Eakins, 2009). The ETOPO1
data has combined the satellite altimeter observations, shipping load sonar measurement, multi
resolutions digital terrain database and the global digital terrain model and many other sources, and been
used in the global and regional oceanic model widely. And the original bathymetry was revised in the
area of next to the coast of China mainland according to the *in-situ* data measured by our group, then
smoothed according to Shapiro (1975). The maximum depth was set to be 6000 m and the minimum
depth to be 10 m in the model (Wang, 1996).
The initial temperature and salinity were derived from the climatology monthly mean Simple Ocean Data
Assimilation (SODA, Carton and Giese, 2008) in January. However, the initial velocities and elevation
were set to zero, which means to integrate the model from a static status. The model's western lateral
boundary was treated as a wall. The other three (northern, southern, eastern) lateral boundaries were
opened, whose temperature, salinity, velocity, and elevation were prescribed by spatial interpolation of
the monthly mean SODA dataset. The 2D and 3D velocities, through the open boundaries, are modulated
to guarantee the conservation of volume flux in the whole model domain. In addition, the nudging
technology was used for 3D velocity, temperature, and salinity to the three open lateral boundaries with a
30-day time scale for outflow and 3-day for inflow.





The model is forced using 6-hourly wind stress, net fresh water flux, net heat flux, surface solar
shortwave radiation at surface from NCEP_Reanalysis 2 data provided by the NOAA/OAR/ESRL PSD,
Boulder, Colorado, USA, from their web site at http://www.esrl.noaa.gov/psd/ (Kanamitsu et al., 2002).
In order to get more reasonable simulated SST, the kinematic surface net heat flux sensitivity to SST
(dQ/dSST) is used to introduce thermal feedback to correct net surface heat flux (Barnier et al., 1995)
with a constant number -30 $W/m^2/℃$ in the whole domain. In addition, the monthly mean climatology
discharges of the Mekong River and the Pearl River are prescribed to the model.
The system was run with 6 seconds time step for the external mode, and 180 seconds for the internal
mode under the initial conditions, boundary conditions and surface forcing mentioned in above. The
system was conducted a hindcast run from 2000 to 2011 after a 15 years climatology run for spin-up
(Wang et al., 2012). The model results are archived to the snapshot with a 5-day interval, which will be
used as the ensemble members for the EnOI (Ensemble Optimal Interpolation) method assimilation.
After the hindcast run, the system was conducted an assimilation run in 2012 with EnOI method, the
along track SLA data from AVISO had been assimilated as the observations with a 7-day time window.
The details on the EnOI applied in the SCSOFS can be referred as Ji et al. (2015). The assimilated results
are archived to daily mean with a 1-day interval in 2012, which will be used to compare and validate in
this paper. Then the system is operating in NMEFC since January 1st, 2013. It runs daily for 6 days
(1-day nowcast and 5-day forecast), and provides 120-hour forecasting products, which including the 3D
ocean temperature, salinity and currents with 24 hours interval.
**2.4 The configurations of MO**
The high resolution global analysis and forecasting system PSY4V1R3 was operational as the V2 of the
MyOcean project from February 2011 up to April 2013, when it was replaced by the PSY4V2R2 system.
During this period, PSY4V1R3 has been producing weekly 14-day hindcasts and daily 7-day forecasts.
The model configuration of PSY4V1R3 is based on a tripolar ORCA grid type (Madec and Imbard, 1996)
in the NEMO 1.09 version with a 1/12 °horizontal resolution which means 9 km at the Equator, 7 km at
mid latitudes and 2km toward the Ross and Weddel Sea. The grid cells follow an Arakawa C-grid type
(Arakawa and Lamb, 1977). The 50-level vertical discretization retained in this system has 1m resolution
at the surface, decreasing to 450m at the bottom and 22 levels within the upper 100m. "Partial cell"
parametrization was chosen for a better representation of the topographic floor (Barnier et al., 2006). The



high frequency gravity waves are filtered out by the free surface formulation of Roullet and Madec

205    (2000).

For the diffusion, a horizontal bilaplacian was added along the equator ($20 m^2 s^{-1}$) and two laplacians in
the Canadian straits (up to $100 m^2 s^{-1}$). Laplacian lateral isopycnal diffusion was added on tracers (125
$m^2 s^{-1}$) and a horizontal biharmonic viscosity was added for the momentum ($-1 \times 1010$ $m^4 s^{-1}$ at the
Equator and decreasing poleward as the cube of the grid size). In addition, the vertical mixing is
parameterized according to a turbulent closure model (TKE order 1.5) adapted by Blanke and Delecluse
(1993), the lateral friction condition is a partial-slip condition with a regionalization for the
Mediterranean Sea, Indonesian region, Canadian straits and Cape Horn. The atmospheric fields are taken
from the ECMWF (European Centre for Medium Range Weather Forecasts) Integrated Forecast System
at a daily average frequency. Momentum and heat turbulent surface fluxes are computed from CLIO bulk
formulae (Goosse et al., 2001). We use a viscous-plastic rheology formulation for the LIM2_VP ice
model (Fichefet and Maqueda, 1997, LIM2_VP in Hunke and Dukowicz 1997,). A multivariate data
assimilation (Kalman Filter kernel with SEEK formulation , Pham et al., 1998) of *in-situ* T and S (from
Coriolis/Ifremer), along track MSLA (from AVISO, with MDT from Rio and Hernandez, 2004) and
intermediate resolution SST (1/4 °SST product RTG from NOAA) is performed with the SAM2 software
(Lellouche et al., 2013). An Incremental Analysis Update (IAU) centered on the 4th day of the 7-day
assimilation window ensures a smooth correction of T, S, U, V and SSH. The assimilation cycle consists
of a first 7-day simulation called guess or forecast, at the end of which the analysis takes place. The IAU
correction is then computed and the model is re-run on the same week, progressively adding the
correction. The increment is distributed in time with a Gaussian shape which is centered on the 4th day.
More details on the SAM2 software (applied on other model configurations) can be found in Lellouche et
al. (2013) except that no large scale bias correction is applied in PSY4V1R3. Concerning the initial
conditions, the PSY4V1R3 was started in April 2009 from a 3D climatology of temperature and salinity
(WOA2005).





### 3 Comparisons, validations and discussion


### 3.1 Velocities


### 3.1.1 Absolute Geostrophic Velocity


Figure 2 shows the distributions of the monthly AGV composited with Sea Surface Height (SSH) from
AVISO, MO, and SCSOFS in January, April, July, and October of 2012, respectively. Here we use the
January, April, July, and October represent winter, spring, summer, and autumn, respectively. It is
valuable to note that the AGV of MO and SCSOFS are not the velocities output from the numerical
model directive. However, in order to better comparison, they are recalculated according to SSH from the
model output on every day and assuming geostrophic balance following Eq. (1):
$$u = -\frac{g}{f}\frac{\partial SSH}{\partial y} \qquad v = \frac{g}{f}\frac{\partial SSH}{\partial x} \qquad\qquad (1)$$
where $g$ is gravitation acceleration, $f$ is the Coriolis parameter, $x$, $y$ are the east, north axis; $u$, $v$ are the
eastward, northward velocity components in horizontal, respectively.
By comparing among the three results, both MO and SCSOFS can catch the main basin-scale oceanic
circulation pattern in the SCS, and show that a cyclonic gyre in winter and an anti-cyclonic gyre in
summer, which being well accordance with the pattern of AVISO, except that the current speeds are a
little stronger than AVISO. It is worth to mention that the result of MO is well agreement with the
AVSIO in January, such as the southward western boundary currents along the eastern coast of Vietnam,
the LCC, the anti-cyclonic eddy in the western of the LUS around (118 °E, 21 °N), the cyclonic eddy in
the eastern of the Vietnam around (113 °E, 15 °N). However, the result of SCSOFS is much smooth
without obvious mesoscale or small scale circulation, or they are very weaker (0.2-0.4m s$^{-1}$) than those
(0.6-0.8m s$^{-1}$) of AVISO or MO. The circulation is chaos in spring in the SCS, though the circulation
pattern of MO is better agreement with the one of AVISO than the one of SCSOFS. All the three results
show the anti-cyclonic eddy around (111 °E, 10 °N) and the western boundary jet in the southeast of the
Vietnam in summer, with the maximum speed being about 1.0m s$^{-1}$, 0.9m s$^{-1}$, and 0.7m s$^{-1}$ for AVISO,
MO, SCSOFS, respectively. The westward intensification along the eastern coast of the Vietnam is most
obvious in autumn than other three seasons, and the maximum speed is more than 1.0m s$^{-1}$ for MO and
SCSOFS, but 0.7m s$^{-1}$ for AVISO.
As mentioned in Sect. 1, the Kuroshio intruding the SCS through the LUS has been distinguished three
types as the looping path, the leaking path and the leaping path, according to Nan et al. (2011a). All three




results show the looping path in winter, the leaping path in summer and leaking path in autumn, which is
well accordance with the model results showed by Wu and Chiang (2007). However, AVISO, MO, and
SCSOFS show the leaking path, looping path, and leaping path in spring, respectively.

**3.1.2 Time series from mooring station**

Figure 3 shows the comparison of the daily mean time series of the u, v components from the mooring,
MO, and SCSOFS in 40m-depth layer at the Maoming station (See Fig. 1) from July 11 to October 8,
2012. Both MO and SCSOFS can catch the same variation trends of the time series with the mooring
observation. Especially, MO results have represented the current variations well for both u- and v-
component, during the period of the Typhoon Kai-tak on 17 August 2012. Although SCSOFS shows the
larger velocity during the Typhoon Kai-tak, the range of large is less than the observation and leading the
observation about 1 day. The root mean square errors (RMSE) between MO and SCSOFS and
observation are 0.075m s$^{-1}$, 0.094m s$^{-1}$ for u-component, 0.062m s$^{-1}$, 0.084m s$^{-1}$ for v-component,
respectively. Overall, MO results are better agreement with the observation than those of SCSOFS.
However, SCSOFS results have a phase bias comparing with the observation, which is leading the
observation about 1 day.

**3.2 Temperature and Salinity**

**3.2.1 SST**

SST is a very important prognostic variable in a hydrostatic ocean general circulation numerical model,
which plays a key role to the ocean circulations and the air-sea interaction. So SST error is crucial criteria
of the numerical model skill, especially for an operational ocean circulation model. In fact, the SST
simulation error is affected by several factors, for example the limitation of physical model, the surface
atmosphere forcing, the bias of initial field and the uncertainty from the open boundary, as pointed out by
Ji et al. (2015). Although the SST data have been assimilated into both MO and SCSOFS, the assimilated
SST still has some errors for both systems.
Figure 4 shows the distributions of the monthly mean SST errors between two systems and MGDSST in
the SCS in 2012. The errors show an obvious regional distribution, the bigger errors mainly exist in the
coastal region for the depth shallower than 200 m, such as in the TWS, the eastern of the Guangdong
province in January, the gulf of Tonkin in July. What's more, the strong seasonal variation for SST error



also can be found, which is larger in winter and smaller in summer, from both systems. Comparing with
MGDSST, the maximum, minimum, and mean monthly RMSE are 0.78℃, 0.37℃, 0.51℃ for the MO,
1.15℃, 0.56℃, 0.86℃ for the SCSOFS, respectively, in the SCS. Based on the Fig. 4, the simulated SST
performance of MO is better than those of SCSOFS by comparing with MGDSST.
**3.2.2 Horizontal and vertical distribution of TS**
The horizontal distributions of 10-m depth layer TS in the eastern of Hainan island from the *in-situ* of
Qiongdong cruise, MO, and SCSOFS, respectively, are shown in Fig. 5. Two clear cold and salty water
cores located at the eastern of Hainan island, which being about (110.75 ℃, 19.2 ℃) and (111.3 ℃,
19.7 ℃), are shown in both *in-situ* and SCSOFS (Fig. 5), except that the SCSOFS being more saline than
*in-situ*. It can be easily deduced that the two cores are produced by upwelling process from the TS
vertical distributions of the section K, F, H, and G (Jing et al., 2015).
Figure 6 shows the vertical TS distributions from the *in-situ* of Qiongdong cruise, MO and SCSOFS,
along section E. Both systems have got the same vertical structures of TS with the *in-situ*. All of them
show out the obvious upwelling system, with cold and salty waters flowing from offshore to nearshore
along the bottom. All three results show the upper mixing layer depth is about 15m, with the sea water
well mixed above 15m depth and the isotherms and isohalines are almost vertical, indicating strong
stratification in summer. The diluted water is flushing from the nearshore to offshore, with the
33-isohaline located at about 50km for both *in-situ* and SCSOFS, but at about 20km for MO. In above, it
is indicated that the results of SCSOFS is better agreement with the *in-situ* than those of MO.
The vertical distributions of TS from the *in-situ* of Nansha cruise, MO, and SCSOFS along the 10 ℃
section are shown in Fig.7 for the layer above 300m and Fig.8 for the layer of 300-1200m. Both systems
have got almost the same vertical structures with the *in-situ*, especially for the upper mixing layer depth
about 70m are shown in the three results. The temperature almost linearly decreases from 28℃ to 3℃
with the depth increasing from the bottom of the upper mixing layer to the 1200m depth. However, the
salinity increases from 33.5 to 34.5 with the depth increasing from the bottom of the upper mixing layer
to about 200m depth, and keeps 34.5 from 200m to 300m depth. Then a fresh water layer exists in the
middle layer from about 400m to 700m with the salinity about 34.4. Below the middle layer, the salinity
again increases from 34.4 to 34.58 with the depth increasing from 700m to 1200m. It indicates that the
results of MO and SCSOFS are well agreement with *in-situ*, except that the salinity of the fresh water in



the middle layer from MO is less than 34.4 and fresher than those of *in-situ* and SCSOFS, but the
thickness of the layer is thicker than those of *in-situ* and SCSOFS.

**3.2.3 Correlation ship between model and *in-situ***

In order to better compare and validate the performances of the two systems, we collected all the
measured TS data from five cruises in the SCS in 2012 to conduct a comprehensive correlation analysis.
Figure 9 shows the comparison of relativity of TS between MO, SCSOFS and *in-situ* by scatter points,
respectively. Any point in the Fig.9 is corresponded with two values of temperature or salinity, one is
from the *in-situ* along X axis, and the other one is from MO or SCSOFS along Y axis. The correlation
coefficients of temperature are 0.987, 0.982, and of salinity are 0.717, 0.897, between MO, SCSOFS and
*in-situ*, over the 95% significance level, respectively, which showing the good relativity between MO,
SCSOFS and *in-situ*. It also indicates that the relativity of temperature is better agreement with *in-situ*
than those of salinity for both MO and SCSOFS, and SCSOFS is better agreement with *in-situ* than MO
for salinity.

**3.3 Mesoscale activities**

**3.3.1 SST front**

Oceanic front is a good indicator for connection between water masses with different hydrological
features, which is an important marine mesoscale phenomenon. There are numerous SST fronts in the
SCS, most of them located on the continental shelf with the depth below 200m or aligned with the shelf
break, especially in the NSCS. A few evident SST fronts have been identified from the long-term
NOAA/NASA Pathfinder SST data, namely: Fujian-Guangdong Coastal Front, Pear River Estuary
Coastal Front, Taiwan Bank Front, Kuroshio Intrusion Front, Hainan Island East Coastal Front, Tonkin
Gulf Coastal Front (Wang et al., 2001). All of them exhibit very strong seasonal variability, which is
mainly due to the EAM (Belkin and Cornillon, 2003).
Figure 10 shows the distributions of SST fronts from MO and SCSOFS for four seasons. The similar
frontal patterns with their evident seasonal variations are shown in both systems, except for some small
differences. In winter, most fronts reach maximum strength (>0.2℃/km). The Fujian-Guangdong coastal
front and Taiwan Bank front are major fronts in the SCS which agree with previous satellite result from
Wang et al. (2001). These two fronts merge and extend to Pearl River Estuary and the Hainan Island. The



Hainan Island East Coastal Front is stronger in MO than in SCSOFS, whereas the Tonkin Gulf Coastal
Front is stronger in SCSOFS than in MO. In SCSOFS, the Kuroshio Intrusion front is obvious, however,
which is hardly seen in MO. In spring, most fronts become weak obviously due to the weakening of
northeast monsoon from both systems, except that the Hainan West Coastal front emerges in SCSOFS. In
summer, weakening almost occurs in all the fronts mentioned above for SCSOFS, which is in agreement
with the result of Wang et al. (2001). However, disappearing occurs in all the fronts for MO. In fall, most
fronts fade and disappear, except that the Taiwan Bank front has very weak strength compared to other
seasons for both systems. Both systems have not shown the Kuroshio Intrusion Front identified by Wang
et al. (2001) in summer and fall.
**3.3.2 The Typhoon Tembin**
There are a lot of typhoons in the SCS during the typhoon season in every year, so that the typhoon
activities are very frequent in the SCS, especially in 2012. One hot study on the air-sea interaction is the
responding of the physical ocean dynamics to typhoon in the oceanic upper layer. One important
responding is the decreasing of SST due to the strong vertical mixing caused by typhoon (Price et al.,
1994). According to the SST observation from the satellite, SST usually decreases 2-5℃ due to typhoon
passing (Cione and Uhlhorn, 2003; D'Asaro et al., 2007; Wu et al., 2008; Jiang et al., 2009). Dare and
Mcbride (2011) researched the response of SST to the global typhoons during 1981~2008 and indicated
that the maximum decreasing of SST usually occurred in 1-day after typhoon passing.
In this section, we select the typhoon Tembin as an example to validate the MO and SCSOFS model skill
for the SST simulation. As shown in Fig. 11, the typhoon Tembin went through and made a perfect turn
around in the NSCS from 25 to 28 August 2012. From the three results, we can find the obvious
decreasing of SST 1-day after typhoon passing, which is about 2-4℃ and well correspondence with
previous studies mentioned in above. SCSOFS is much better agreement with OISST than MO,
especially on 26 and 27 August 2012, not only for the range of SST decreasing, but also for the domain of
SST deceasing.
**3.3.3 Mesoscale eddy**
Mesoscale eddies cannot be identified and extracted from geophysical turbulent flow as observed by
satellite altimetry without suitable definition and a competitive identification algorithm. A multitude of



different techniques for automatic identification of eddies have been proposed based either on physical or
geometric criteria of the flow field. In this study, a free-threshold eddy identification algorithm with the
SLA data is employed. This algorithm is based on the vector geometry method and Okubo-Weiss method
(Okubo, 1970; ) with six constraints applied to the SLA to detect an eddy: (1) a vorticity-dominated
region at the eddy center ($W < 0$) must exist; (2) the SLA magnitude has a local extreme value (minimum
or maximum); (3) closed contours of SLA around the eddy center must exist; (4) the eddy radius must be
larger than 45km.(5)the eddy amplitude must be larger than 4cm. In this study, the amplitude is defined
as the absolute value of the SLA difference between the eddy center and the SLA along the eddy edge.
The Eddy-tracking method used is the one developed by Chaigneau et al. (2008), and we only keep
eddies with life span not less than 28 days. Eddies were analyzed and compared based on MO, AVISO
and SCSOFS in 2012. The numbers of eddies for three types of data were in Table 1, cyclones and
anti-cyclones were counted separately and seasonally.
The spatial distribution of eddy birthplace is shown in Fig 12. MO has more eddies formed, especially
anti-cyclones formed than those of AVISO, most of the excessive eddy cores were found near the middle
of SCS. SCSOFS has more anti-cyclones as well and less cyclones than AVISO. Both MO and SCSOFS
show excessive eddies formed in the middle of the basin and less eddies in the western of the east of
Vietnam. The SLA of SCSOFS and MO is calculated simply by subtracting mean SSH (24 years mean
for SCSOFS and only one-year mean for MO) instead of an uniformed Mean Sea Surface, which might
cause the excessive anti-cyclones in both models. All three types of data agree that less eddies formed in
the middle part of NSCS.
As for the seasonal distributions (figures not shown), all three data have most eddies in spring and lest in
fall. Both AVISO and SCSOFS have more cyclones than anti-cyclones in spring and fall, and all three
have less cyclones in summer. SCSOFS differs with AVISO mainly in winter while they agree
reasonably in the other three seasons. MO has surplus eddies counted in every season especially for
anti-cyclones, which might because of the error introduced by the simplified calculation of SLA.
**4 Conclusions**
Two operational ocean analysis and forecasting systems, MO and SCSOFS, have been built based on the
state-of-the-art hydrodynamic ocean model in France and China, respectively. The comparison and





validation for the performance of both systems on the ocean circulation, the structures of the TS, and
mesoscale activities in the SCS, based on the observed satellite and *in-situ* data in 2012, are shown in this
paper. The comprehensive performances for the both systems are summarized as follow.
Both systems have caught the main basin-scale circulations in the SCS and been well agreement with the
result of AVISO. And the results of MO are better agreement with those of AVISO than those of
SCSOFS for several branches and eddies in January. There are no many mesoscale or small scale
circulations shown in SCSOFS, which may because of a little strong horizontal mixing set in the model.
The westward intensification in the eastern coast of the Vietnam is most strong in autumn among the four
seasons. For the type of the Kuroshio intruding the SCS, the three results show the looping path in winter,
the leaping path in summer and leaking path in autumn. However, the leaking path, looping path and
leaping path are shown for AVISO, MO and SCSOFS in spring, respectively.
Both systems get the same variation of the u-/v- components time series with the mooring observation.
The RMSE between MO, SCSOFS and mooring observation are 0.075m/s, 0.094m/s for u-component,
0.062m/s, 0.084m/s for v-component, respectively. The results of MO are better agreement with the
observation than those of SCSOFS, especially during the period of the Typhoon Kai-tak.
The maximum, minimum, and mean monthly RMSE between MO and MGDSST are 0.78℃, 0.37℃,
0.51℃, between SCSOFS and MGDSST are 1.15℃, 0.56℃, 0.86℃ for the SCSOFS in the SCS,
respectively. For the horizontal and vertical distributions of TS, both systems have got the same
structures with the *in-situ*, but the results of SCSOFS are better agreement with the *in-situ* than those of
MO. The correlation coefficients of temperature are 0.987, 0.982, and of salinity are 0.717, 0.897,
between MO, SCSOFS and *in-situ*, over the 95% significance level, respectively. It indicates that the
good relativity between MO, SCSOFS and *in-situ*, the relativity of temperature is better agreement with
*in-situ* than those of salinity for both MO and SCSOFS, and SCSOFS is better agreement with *in-situ*
than MO for salinity.
The similar SST frontal patterns with their evident seasonal variations are shown in both systems. Most
fronts achieve maximum strength in winter, become weak obviously due to the weakening of northeast
monsoon EAM in spring and summer, fade and disappear in autumn. It is well agreement with the result
of Wang et al. (2001).
During the typhoon Tembin in the NSCS, the obvious decreasing of SST about 2-4℃ occurs 1-day after
typhoon passing shown in the results of MO, SCSOFS and OISST, which is well agreement with





previous studies. SCSOFS is much better agreement with OISST than MO both for the range and domain
of SST decreasing.
MO has more eddies formed near the middle of SCS than AVISO, especially for anti-cyclones. SCSOFS
has more anti-cyclones as well, but less cyclones than AVISO. All three data have most eddies in spring
and lest in fall, and less cyclones than anti-cyclones in summer. Both AVISO and SCSOFS have more
cyclones than anti-cyclones in spring and fall.
In order to improve their performances further in the SCS, according to the comparison and validation for
the two systems, MO and SCSOFS, we would like to propose some suggestions to modify the systems.
For MO, we would like to suggest (1) to modify the model bathymetry in the coast area for the depth less
than 200m to improve the model performance in shallow water area, such as SST front; (2) to change the
initial conditions of TS to improve the TS vertical structures, especially for the salinity in deep water area;
For SCSOFS, we would like to suggest (1) to weaken horizontal mixing to get more reasonable
mesoscale or small scale circulations; (2) to optimize the data assimilation scheme further to better
assimilate the *in-situ* and satellite data; (3) to replace the surface forcing data with the higher horizontal
or temporal resolution; (4) to replace the boundary conditions from monthly to weekly or daily, such as
MO. For both systems, we also would like to suggest to try to get and assimilate more observed data
during the typhoon period to catch the typhoon process more exactly.
**Author contribution**
X. Zhu, H. Wang and G. Liu compared and validated the model results on velocities and TS. C.
Régnier and M. Drévillon build the MO, D. Wang build the SCSOFS. X. Kuang analyzed the model
results on mesoscale eddy. S. Ren analyzed the model results on SST front. Z. Jing provided the *in-situ*
data. X. Zhu prepared the manuscript with contributions from all co-authors.
**Acknowledgements**
We would like to thank the anonymous reviewers and the Editor,    , for their valuable contributions
that allow us to improve the manuscript substantially. This study is supported by the National Natural
Science Foundation of China under contract No. 41222038, 41376016, 41206023 and the
Strategic Priority Research Program of the Chinese Academy of Sciences Grant No. XDA1102010403.



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






Table 1 Eddy Numbers of different datatype

| | AVISO | | | MO | | | SCSOFS | | |
|---|---|---|---|---|---|---|---|---|---|
| | CYCL | ACYCL | TOTAL | CYCL | ACYCL | TOTAL | CYCL | ACYCL | TOTAL |
| Spring | 6 | 3 | 9 | 6 | 7 | 13 | 6 | 3 | 9 |
| Summer | 2 | 3 | 5 | 4 | 7 | 11 | 3 | 5 | 8 |
| Fall | 2 | 1 | 3 | 6 | 7 | 13 | 2 | 1 | 3 |
| Winter | 5 | 2 | 7 | 5 | 5 | 10 | 1 | 3 | 4 |
| Overall | 15 | 9 | 24 | 21 | 26 | 47 | 12 | 12 | 24 |






Figure 1: The model domain and bathymetry of SCSOFS.







Figure 2: The monthly mean Sea Surface Height (color shaded) and the corresponding surface absolute geostrophic velocity (units: m s$^{-1}$) in January, April, July, and October, 2012. The left panels are from AVISO, the middle panels are from Mercator Océan, the right panels are from SCSOFS.



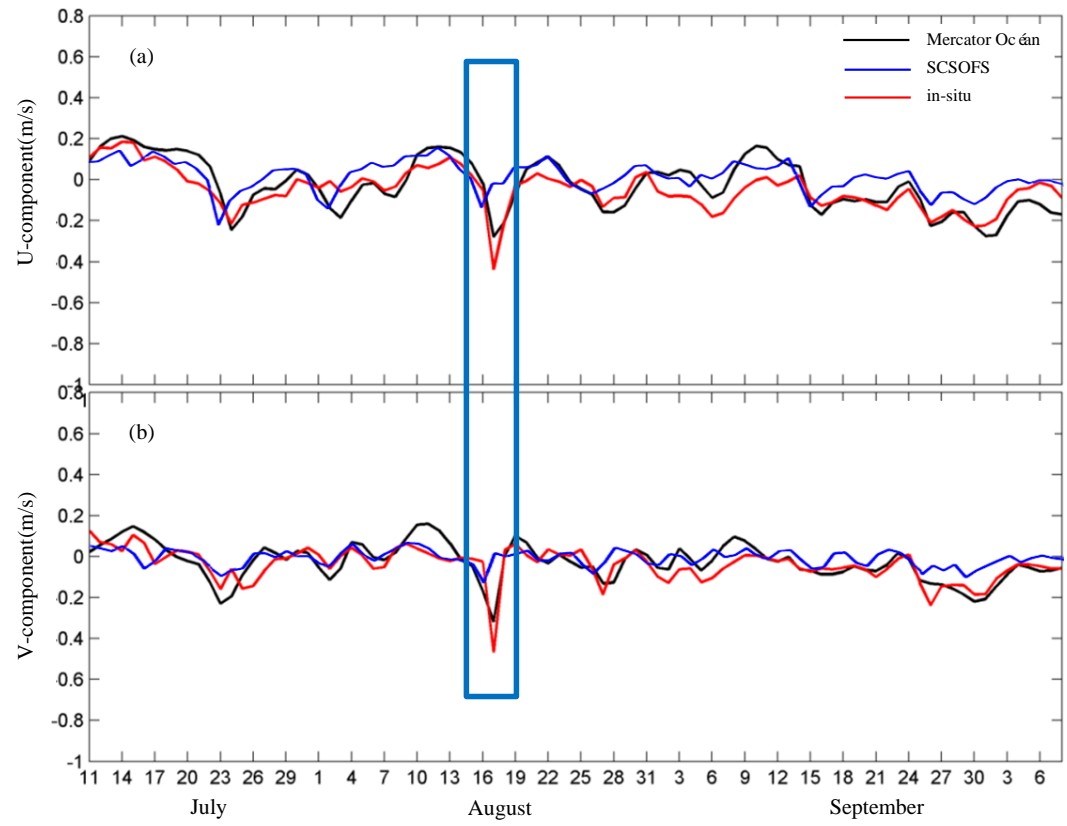

Figure 3: The daily mean time series of u(a) and v(b) components in 40m-depth layer at the Maoming mooring station, from in-situ, Mercator Océan, and SCSOFS.





Figure 4: The monthly mean SST error between Mercator Océan (left panels), SCSOFS (right panels) and MGDSST in January, April, July, and October, 2012





Figure 5: The horizontal distributions of temperature (upper panels) and salinity (lower panels) at 10-m depth layer from the *in-situ* observations of Qiongdong cruise (left column), Mercator Océan (middle column), and SCSOFS (right column), respectively.

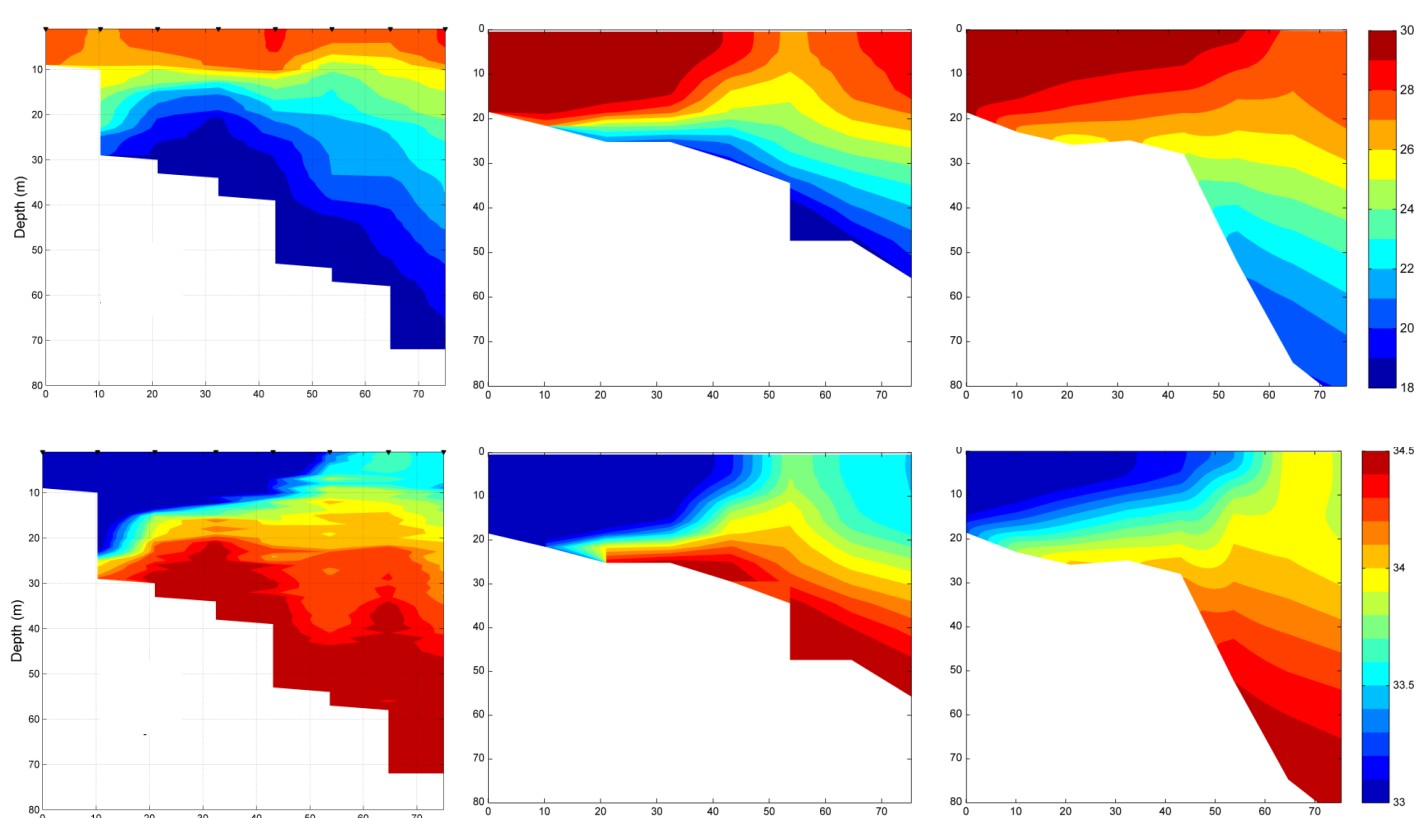

Figure 6: The vertical distributions of temperature (upper panels) and salinity (lower panels) along the section E (See Fig.1) from the *in-situ* observations of Qiongdong cruise (left column), Mercator Océan (middle column), and SCSOFS (right column), respectively.





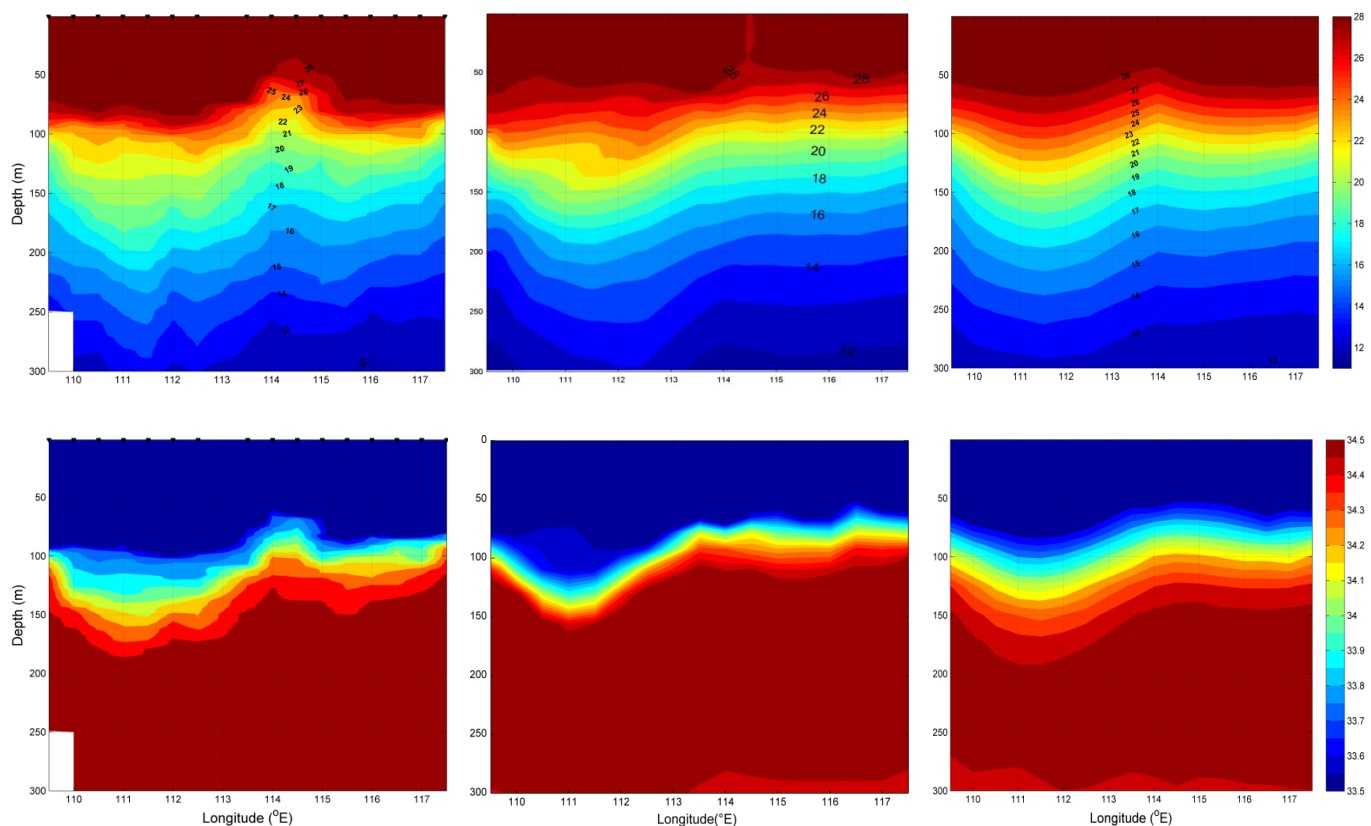

Figure 7: The vertical distributions of temperature (upper panels) and salinity (lower panels) in above 300m depth along the section 10° N from the *in-situ* observations of Nansha cruise (left column), Mercator Océan (middle column), and SCSOFS (right column), respectively.





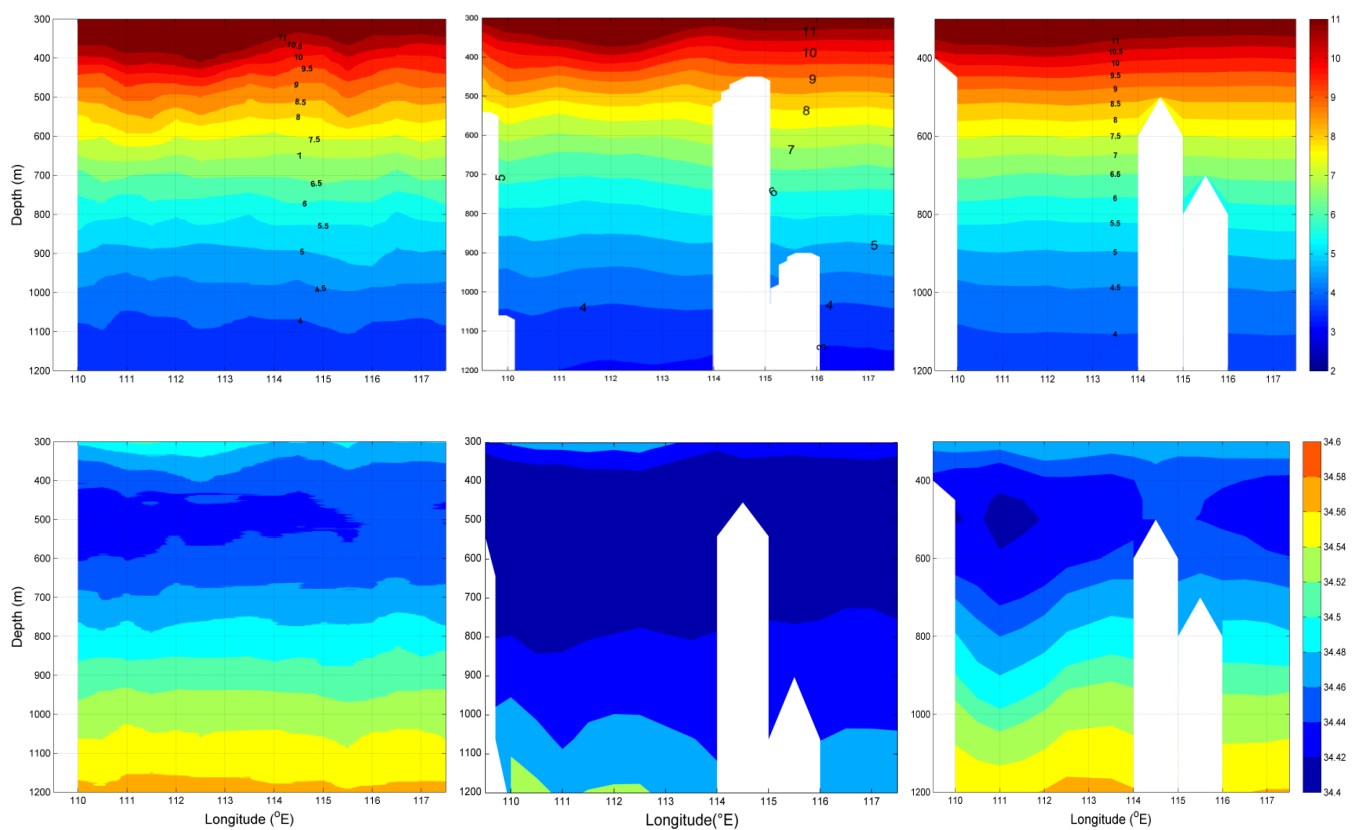

Figure 8: The same with Fig.7, but for the deep layer with depth from 300m to 1200m.




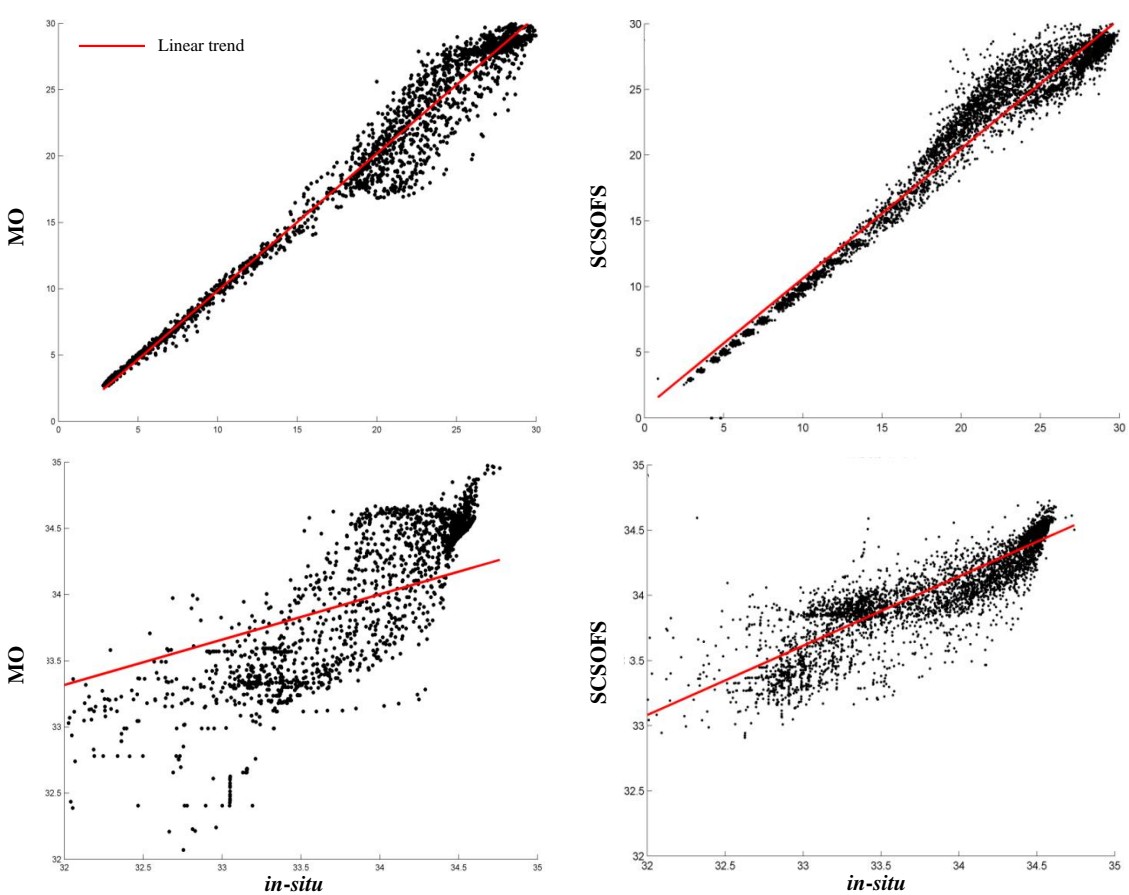

Figure 9: The relative relationships of temperature (upper panels) and salinity (lower panels) between Mercator Océan (left column), SCSOFS (right column) and the *in-situ* observations of all cruises.





Figure 10: The distributions of SST fronts in the SCS from Mercator Océan (left panels) and SCSOFS (right panels) in January, April, July, and October.




Figure 11: The SST differences of the day from the last day during the period of Typhoon Tembin. The
black dots are the positions of the Typhoon Tembin at 00h, 06h, 12h, and 18h on each day.





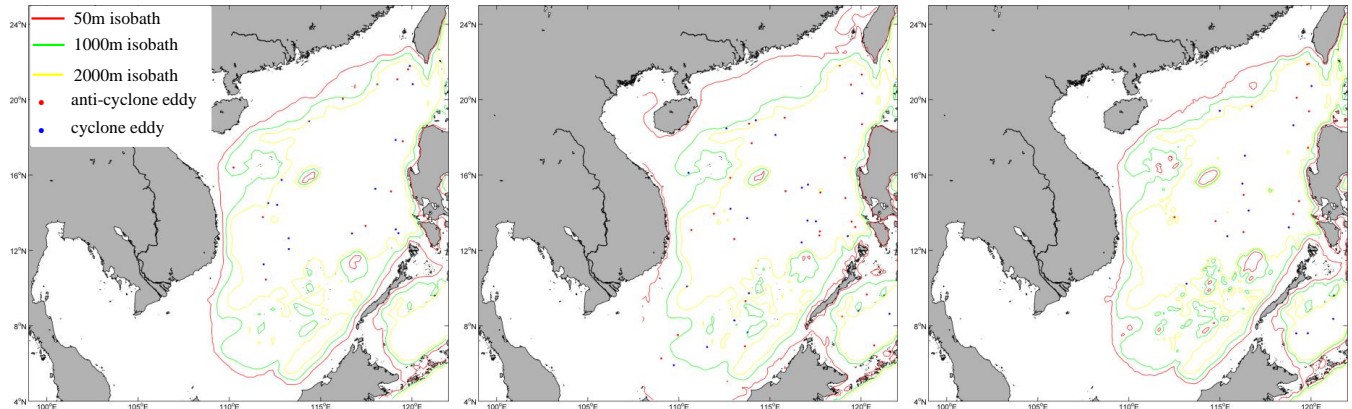

Figure 12: The spatial distributions of eddy birthplace identified using the method of Chaigneau et al. (2008) in the SCS from AVISO (left), Mercator Océan (middle) and SCSOFS (right) in 2012.