# Peer review of "Comparison and validation of global and regional ocean"

_Natural Hazards and Earth System Sciences, 2016_

## Referee Comment (RC1) · L. Lusito (Referee) · 31 Mar 2016

**1. General comments**

First of all, I would like to congratulate all the authors for the good scientific level of the work presented in this paper.

This paper addresses the relevant problem of comparison and validation with real observations of two ocean models (SCSOFS and MO) in the South China Sea, an area which is becoming more and more strategic. Ocean models, in general, are the key to improve our knowledge of the sea state, present and future, on which to base political, environmental, economical decisions by governments and other stakeholders. In this respect, this issues are relevant not only to the general themes addressed by this journal, but also they are especially relevant in the context of this special issue subject

'Situational sea awareness technologies for maritime safety and marine environment protection'.

The results described here can be considered innovative; in fact, even if the South Cina Sea Ocean Forecasting System for the regional modelling of the south China Sea, and the global Ocean model, developed by Mercator Ocean, have been described in previous works, their comparison and validation with real observations in the South China Sea is a new important result that will allow a substantial improvements of the ocean forecasting capability in the area. Moreover the paper shows how both models could be improved regarding the forecast skills in case of devastating events like typhoons which are unfortunately not rare in that region.

The title is clear and reflects the content of the paper. The abstract provides a good summary of the results and of the conclusions that derive from them in a language easily understandable by the average reader. Every section describes clearly a particular aspect of the data, model or method used. The overall length is adequate and a good compromise between a a too-long discussion containing a full description of many technical details and a too short description of the results without a proper introduction of the context, methods and data used.

It is strongly recommended that the English language is revised and improved by a native speaker although the actual form is sufficiently understandable. Some suggestions to improve the text will be reported separately.

Other comments (about units, figures, references, further clarifications) are reported in the next sections.

2. Specific scientific comments

Line 17: explain or define better what you are referring to with the term "mesoscale activities"

Line 74: explain better what you mean with the phrase "where they can then impact

budgets of the tracers"

Line 107: if the satellite data you use are from 2012, what is "April 2014" referring to? The starting date of the SSALTO/DUACS new data system?

Line 109: quoting from the user manual of the L4 altimeter data you are using (and that you cite http://www.aviso.altimetry.fr/fileadmin/documents/data/tools/hdbk_duacs.pdf) "Change of resolution: in DUACS 2014 version, after the feedback from users, the Mercator grid projection with 1/3 x1/3 spatial resolution (Global product) is abandoned. The DUACS 2014 Global products are directly computed on a Cartesian 1/4 x1/4 spatial resolution." Therefore, depending which data you are using, it may be incorrect saying that the products are sampled from the Mercator gridded data so please, specify better if you are using the DUACS 2014 data or a former version (and which one). From the following discussion it seems you are using the DUACS 2014 data.

Line 115: quoting the JMA database description of the MGDSST (http://near-goos1.jodc.go.jp/rdmdb/format/JMA/mgdsst.txt): "Merged satellite and in-situ data Global Daily Sea Surface Temperature (MGDSST): The MGDSSTs are analysed at the Office of Marine Prediction of the JMA with 1/4-degree grid resolution on the near-real-time basis. SSTs derived from satellite's infrared sensors (AVHRR/NOAA) and microwave sensor (AMSR-E/AQUA), and in-situ SST (buoy and ship) are used in the analysis." The list of satellite products here does not match the list of satellite products you mention in the paper

Line 122-127 please clarify in the paper that the reason for which you are using the "AVHRR-only" data is because the production of the AVHRR+AMSR data ended in 2011. Otherwise questions might arise on the impact of the usage of "AVHRR+AMSR" data in your analysis and how the results might differ from the results using "AVHRR-only" data.

Line 137-138: just for my education: why to filter out the tidal signal you use a period of 25 hours and not, for example 24 hours? Same, why did you use a 25-hour period

to calculate the daily average?

Line 140 and following: you mention that there were 5 cruises to measure the temperature/salinity data but you use data from only two of them for the TS distribution comparison? Why only two? Why do you use the data from these particular cruises? Are the data from the remaining three cruises significantly different in some ways? What changes/improvements/impacts do you expect when using all the data available?

Line 244: be more specific than "little stronger than AVISO" For example you can add a time-series-like plot showing the basin-averaged, minimum and maximum velocities (in separated components u and v) for the two models and the satellite observations in the four months. On the x-axis you have the 4 time steps (Jan, Apr, Jul, Oct.), on the y-axis 3 u-velocities (min, max, basin-averaged) for each of the two models and the observation. Same for v. Alternatively, at least quote the basin-averaged velocities in the text. From the Figure only, it is very difficult to distinguish the length of the vectors hence to have an estimation of the magnitude of the velocities. Moreover, while on lines 243 you say that AVISO shows currents that are smaller than MO or SCSOFS, at lines 249 you say that AVISO has comparable velocities to MO and SCSOFS has smaller velocities so there is an incongruence in the text.

Line 260: explain why in your opinion in spring the two models and the observation show each a different type of Kuroshio intrusion. Is this maybe due to some physics effects modelled differently in the two models or boundary conditions not implemented in the best way in the two models. Can you also explain why this effect is visible mainly in spring?

Line 271: i would prefer to use the word "temporal" instead of "phase" bias in this case. The word phase is generally more used when speaking about angles. Do you have an explanation about this temporal behaviour of SCSOFS?

At line 281 you say that the SST has been assimilated in SCSOFS but this is not mentioned when you describe which data are assimilated in the model in lines 187 and

188

At line 286 you say that the SST variation is larger in winter and smaller in summer for both MO and SCSOFS but when looking at Figure 4 this appears to not be true: the absolute magnitude of the variation is much larger for SCSOFS (where you see large blue and red areas) than for MO (where you see prevalent green everywhere) and has the same values (but with opposite signs) for summer and winter; in winter the area that show a large variation is simply larger so it is better to specify the basin-averaged variation is larger. What it seems more correct is that SCSOFS overestimates the SST in winter and underestimates the SST in summer, therefore has a more uniform trend of the SST through the year with respect to MO

Line 287 you quote 3 values for RMSE for each of the models saying that they are the maximum minimum mean monthly but it is no clear what do you mean with "monthly": are these the averaged values over the 4 months or do they correspond to a specific month? In any case please quote the values for each month for a better comparison of the performances of MO and SCSOFS

Line 303: you say that the isohaline is located at 50 km for in-situ data and SCSOFS and at 20 km for MO but this big difference is not so evident in Figure 6 , therefore from Figure 6 it cannot be concluded that SCSOFS performs better than MO when compared to in-situ data. A plot of a vertical profile of TS and the TS bias for let's say 20km, 50 km and 70 km can clarify better the performances of the models in this case.

Line 375 Define what is W

Line 387 Why do you use a period of 24 years for SCSOFS and just one year for MO?

Line 391 You say that all the three results show a small number of eddies in autumn and a larger number of eddies in spring but this is not true for MO when looking at table 1, where MO predicts 13 total eddies for both spring and autumn

Line 395 explain better the oversimplification of SLA calculation for MO and why SC-

[Figure]

SOFS does not suffer from this.

3. Style comments and suggestions

3.1 General

Be consistent with the space between the value and the unit of measure, for example in line 28 you write "1200m" and line 156 you write "0.16 m". As a general reference, the NIST Guide for the Use of the International System of Units (SI) states "7.2 Space between numerical value and unit symbol: In the expression for the value of a quantity, the unit symbol is placed after the numerical value and a space is left between the numerical value and the unit symbol. The only exceptions to this rule are for the unit symbols for degree, minute, and second for plane angle (...) in which case no space is left between the numerical value and the unit symbol."

Change all the "northeastly" in "northeasterly" , "southwestly" in "southwesterly" and all the similar words

Use consistently "coastal currents" or "Coastal Currents" (check for examples line 50-51)

Be consistent in using 1/4 or 0.25 for the horizontal resolutions (for example lines 116 and 119)

In the section about MO (2.4), you keep calling the model PSY4V1R3 systematically without any mention of Mercator Ocean and in the later sections this name (PSY4V1R3) disappears. Please introduce at some point in section 2.4 a clarification like : the PSY4V1R3 configuration described here is indicated for as MO model through this paper.

3.2 References

Bell, 2015 is never used Chu, 2001 is never used Daudin, 2013 never used Weiss 1991, not used Line 167: move the reference to SODA to the line where you first talk

about SODA, i.e. line 91; Line 179: The reference for Barnier 1995 is missing. In the reference there is a Barnier 2006. Please check. Line 185: the reference for Wang et al 2012 is missing. Please check Line 201: move the reference to the Arakawa C-grid to where you first introduce the Arakawa C-grid, i.e. line 152. Line 228 : the reference WOA2005 is missing

3.3 Figures

In general increase the size of the x/y labels especially in the maps and use higher resolution files so that the image does not loose sharpness when zooming in

Fig 1: change the colours for the cruises paths and the mooring station because now they cannot be easily distinguished from the background Moreover, please indicate in Fig.1 more of the channels and seas you name in lines 30-33 to facilitate the non-expert readers in understanding the unique geographical features of the SCS. Reduce the width (or the size in general to keep the aspect ratio) because the label of the scale is outside the printing area so it is missing in a printed version of the paper

Fig.2: report in a separate plot the mean maximum and minimum AGV because it is currently difficult to compare them from the maps shown in Fig.2 or report these values explicitly in the text. Increase the sizes of the labels on the legend and axes. Use higher resolution files. The unity of measure is missing from the scale

Also SSH bias maps can be added (MO minus AVISO and SCSOFS minus AVISO) to evidence better the behaviour of the two models with respect to the observations (analogous to Fig.4 maps)

Maybe you can also change the color map for Figure 4: a red/white/blue (RWB) map is usually more appropriate to represent bias. For example in the actual maps the green color can correspond to bias values of both +0.5 and -0.5; using a RWB map would make more clear the areas where the bias is positive and where it is negative.

Fig.5-6-7-8: units missing from the colorbar

Fig.9: the correlation plots for salinity show a less good linear relationship between MO and SCSOFS and data with respect to the same plots for temperature? Did you try to plot the correlations in different depth ranges to see if the plots show a better linear correlation and if there is a depth range for which the correlation is not good and this degrades the overall linear relationship?

Fig 10: change the colormap so that also the pale yellow structures can be distinguished better from the white background

Fig.11: explain what are the white areas on the map

3.4 Language and typos

Line 12-13: change "Mercator Ocean...in China" in "the global Mercator Ocean Operational System , developed and maintained by Mercator Ocean in France and the regional South China Sea Operational Forecasting System (SCSOFS) by the National Marine Environmental Forecasting Center (NMEFC) in China". I think it is better to underline that MO is a global ocean forecasting system developed by Mercator Ocean, a scientific institution in France.

Line 22: change "AVISO data" in "satellite observations": at this point it is not yet clear to a medium reader what are AVISO data; change "results compared in above" in "outcome of the results comparison"

Line 42-43: change "in the NSCS....in the SSCS" in "is present in the NSCS, while a semiannual/biennial change from a cyclonic gyre regime in winter to an anti-cyclonic gyre regime in summer can be observed in the SSCS" Please note that the word "biannual" is ambiguous; some use it with the meaning of "twice per year" in which case it is better to use the word "semiannual", others use it to say "every two years", in which case it is better to say "biennial".

Line 136 change "abnormal" in "outlier"

Line 142/263: change "See" in "see"

Line 145-147: change "All the measured...correlation analysis" in "The TS data collected from all the 5 cruises will be used to perform a correlation analysis of each of the simulated predictions of MO and SCSOFS models with the observations."

Line 149: change "Ocean" in "Oceanic" and modelling in "modeling". The first correction comes from the original name of ROMS while the second is to align the spelling of the word to the American English rule that you are using in other words (for example as in "analyzed")

Line 208: the value "-1x1010 mˆ4sˆ-1" seems odd, please check in case there is a typo.

Line 216: remove the comma before the parenthesis

Line 218 add a "-" between along and track

Line 221: since it is the first time you mention SSH add the full name: Sea Surface Height (SSH) and remove the full name from line 232 line 267 there must be a typo in "the range of large is less than". Please rewrite. Change "leading" in "anticipating"

line 317 change "ship" in "analysis"

line 320 remove "of relativity"

line 354 substitute "hot" with important

line 367 substitute "SST deceasing" with "SST decrease"

line 415 there is a typo: the first SCSOFS should be changed in MO

---

## Author Comment (AC1) · 20 Apr 2016

We sincerely appreciate the reviewer for the constructive comments and suggestions. We addressed these concerns in the revision by following all the suggestions to significantly improve the manuscript in the following manner: 1) We have explained relative words or sentences and answered all the questions of the reviewer. 2) We have clarified the introductions to the satellite data, SLA, MGDSST, AVHRR in section 2.1. 3) We redraw or modified relative figures mentioned by the review, such as Fig. 1,2,4,5,6,7,8,10. 4) We revised all the styles, language, typos, references 2. Specific scientific comments Line 17: explain or define better what you are referring to with the term "mesoscale activities" We have explained it by adding the words ", such as ocean fronts, Typhoon, and mesoscale eddy," Line 74: explain better what you mean with the phrase "where they can then impact budgets of the tracers"

We have changed it to "where they can then impact budgets of heat, mass, momentum and biogeochemical properties". Line 107: if the satellite data you use are from 2012, what is "April 2014" referring to? The starting date of the SSALTO/DUACS new data system? The satellite datasets, DUACS 2014 version, are distributed to users in April 2014, which including the data from 1993 to present. We only use the data of 2012 to compare and validate our systems' results in this paper. Line 109: quoting from the user manual of the L4 altimeter data you are using (and that you cite http://www.aviso.altimetry.fr/fileadmin/documents/data/tools/hdbk_duacs.pdf) "Change of resolution: in DUACS 2014 version, after the feedback from users, the Mercator grid projection with 1/3 x1/3 spatial resolution (Global product) is abandoned. The DUACS 2014 Global products are directly computed on a Cartesian 1/4 x1/4 spatial resolution." Therefore, depending which data you are using, it may be incorrect saying that the products are sampled from the Mercator gridded data so please, specify better if you are using the DUACS 2014 data or a former version (and which one). From the following discussion it seems you are using the DUACS 2014 data. Yes, we use the DUACS 2014 data in this paper and have modified it in the revised manuscript. Line 115: quoting the JMA database description of the MGDSST (http://neargoos1. jodc.go.jp/rdmdb/format/JMA/mgdsst.txt): "Merged satellite and in-situ data Global Daily Sea Surface Temperature (MGDSST): The MGDSSTs are analysed at the Office of Marine Prediction of the JMA with 1/4-degree grid resolution on the near realtime basis. SSTs derived from satellite's infrared sensors (AVHRR/NOAA) and microwave sensor (AMSR-E/AQUA), and in-situ SST (buoy and ship) are used in the analysis." The list of satellite products here does not match the list of satellite products you mention in the paper. Thanks, we have revised it to match the description of the MGDSST in the website. Line 122-127 please clarify in the paper that the reason for which you are using the "AVHRR-only" data is because the production of the AVHRR+AMSR data ended in 2011. Otherwise questions might arise on the impact of the usage of "AVHRR+AMSR" data in your analysis and how the results might differ from the results using "AVHRRonly" data. Thanks, we have clarified it in the revised manuscript by adding the words "Since the production of the AVHRR+AMSR data ended in 2011" at line 145. Line 137-138: just for my education: why to filter out the tidal signal you use a period of 25 hours and not, for example 24 hours? Same, why did you use a 25-hour period to calculate the daily average? It is because the periods of some of the major diurnal tidal constituents are longer than 24 hours. Actually 26 hours is better than 25 hours to filter out the tidal information. Line 140 and following: you mention that there were 5 cruises to measure the temperature/salinity data but you use data from only two of them for the TS distribution comparison? Why only two? Why do you use the data from these particular cruises? Are the data from the remaining three cruises significantly different in some ways? What changes/improvements/impacts do you expect when using all the data available? We did have compared both two systems with other three cruises, the results are almost the same. In order to save space, we only show two of them in this paper. The Qiongdong cruise is conducted in the coastal area, the Nansha cruise is conducted in the deep area. We also compared all the measured TS data from 5 cruises with the two system, please see Fig. 9 and section 3.2.3 Correlation ship between model and in-situ. Line 244: be more specific than "little stronger than AVISO" For example you can add a time-series-like plot showing the basin-averaged, minimum and maximum velocities (in separated components u and v) for the two models and the satellite observations in the four months. On the x-axis you have the 4 time steps (Jan, Apr, Jul, Oct.), on the y-axis 3 u-velocities (min, max, basin-averaged) for each of the two models and the observation. Same for v. Alternatively, at least quote the basin-averaged velocities in the text. From the Figure only, it is very difficult to distinguish the length of the vectors hence to have an estimation of the magnitude of the velocities. Moreover, while on lines 243 you say that AVISO shows currents that are smaller than MO or SCSOFS, at lines 249 you say that AVISO has comparable velocities to MO and SCSOFS has smaller velocities so there is an incongruence in the text. Thanks. It is not correct, we have deleted the words "except that the current speeds are a little stronger than AVISO". Line 260: explain why in your opinion in spring the two models and the observation show each a different type of Kuroshio intrusion.

Is this maybe due to some physics effects modelled differently in the two models or boundary conditions not implemented in the best way in the two models. Can you also explain why this effect is visible mainly in spring? It is obviously the types of Kuroshio intrusion are different from each among the three results, you can refer the paper Nan et al. (2011a). And AVISO, MO, and SCSOFS show the leaking path, looping path, and leaping path in spring, respectively. Actually, I have not researched much on this problem. I think it is an interesting scientific problem, and additional work need to do to study it in detail. But this is out of the scope of this paper. In my opinion, it may due to the surface wind forcing are different for the two models. Since the wind is weaker in spring than other seasons in this area, the wind forcing used in both two models may not agreement well with the real wind. Line 271: I would prefer to use the word "temporal" instead of "phase" bias in this case. The word phase is generally more used when speaking about angles. Do you have an explanation about this temporal behaviour of SCSOFS? Thanks, we have changed the word from "phase" to "temporal". It may due to the surface wind forcing, we will double check about it. At line 281 you say that the SST has been assimilated in SCSOFS but this is not mentioned when you describe which data are assimilated in the model in lines 187 and 188. Yes, the SST data has been assimilated in SCSOFS by the thermal feedback method to correct net surface heat flux (Barnier et al., 1995). We have modified the description at Line 189 and 190. At line 286 you say that the SST variation is larger in winter and smaller in summer for both MO and SCSOFS but when looking at Figure 4 this appears to not be true: the absolute magnitude of the variation is much larger for SCSOFS (where you see large blue and red areas) than for MO (where you see prevalent green everywhere) and has the same values (but with opposite signs) for summer and winter; in winter the area that show a large variation is simply larger so it is better to specify the basin-averaged variation is larger. What it seems more correct is that SCSOFS overestimates the SST in winter and underestimates the SST in summer, therefore has a more uniform trend of the SST through the year with respect to MO Thanks, we have modified it at line 286. Line 287 you quote 3 values for RMSE for each of the models saying that they are the maximum minimum mean monthly but it is no clear what do you mean with "monthly": are these the averaged values over the 4 months or do they correspond to a specific month? In any case please quote the values for each month for a better comparison of the performances of MO and SCSOFS Thanks, we have modified it at line 298 in the revised version. Line 303: you say that the isohaline is located at 50 km for in-situ data and SCSOFS and at 20 km for MO but this big difference is not so evident in Figure 6, therefore from Figure 6 it cannot be concluded that SCSOFS performs better than MO when compared to in-situ data. A plot of a vertical profile of TS and the TS bias for let's say 20km, 50 km and 70 km can clarify better the performances of the models in this case. We have changed the plot of SCSOFS in Figure 6, and revised the position of MO from 20km to 40km. Line 375 Define what is W W is the Okubo-Weiss parameter (Xiu et al., 2010) defined as: ïïjŇ Where

Line 387 Why do you use a period of 24 years for SCSOFS and just one year for MO? Since we have got only the year of 2012 data from MO, only one year of MO data is available in this study, and 14 years, from 2001 to 2014 for SCSOFS. Line 391 You say that all the three results show a small number of eddies in autumn and a larger number of eddies in spring but this is not true for MO when looking at table 1, where MO predicts 13 total eddies for both spring and autumn Thanks, we have revised it as:" As for the seasonal distributions (figures not shown), all three data have most eddies in spring. Both AVISO and SCSOFS have lest eddies in fall and more cyclones than anti-cyclones in spring and fall, and all three have less cyclones than anti-cyclones in summer." Line 395 explain better the oversimplification of SLA calculation for MO and why SCSOFS does not suffer from this. Oversimplification hear means SLA calculated by only one year's MO data averaged and extracted from SSH. This may introduce great error for SLA and the eddy identification. 3. Style comments and suggestions 3.1 General Be consistent with the space between the value and the unit of measure, for example in line 28 you write "1200m" and line 156 you write "0.16 m". As a general reference, the NIST Guide for the Use of the International System of Units (SI) states "7.2 Space between numerical value and unit symbol: In the expression for the value of a quantity, the unit symbol is placed after the numerical value and a space is left between the numerical value and the unit symbol. The only exceptions to this rule are for the unit symbols for degree, minute, and second for plane angle (...) in which case no space is left between the numerical value and the unit symbol." Thanks, we have modified all the expressions in the revised paper. Change all the "northeastly" in "northeasterly" , "southwestly" in "southwesterly" and all the similar words. Thanks, we have changed them in the revised paper. Use consistently "coastal currents" or "Coastal Currents" (check for examples line 50-51) Thanks, we have changed them in the revised paper. Be consistent in using 1/4 or 0.25 for the horizontal resolutions (for example lines 116 and 119) Thanks, we have changed them in the revised paper. In the section about MO (2.4), you keep calling the model PSY4V1R3 systematically without any mention of Mercator Ocean and in the later sections this name (PSY4V1R3) disappears. Please introduce at some point in section 2.4 a clarification like : the PSY4V1R3 configuration described here is indicated for as MO model through this paper. Thanks, we have clarified it in the revised paper. 3.2 References Bell, 2015 is never used; Chu, 2001 is never used; Daudin, 2013 never used; Thanks, we have deleted these references. Weiss 1991, not used; Thanks, we have added the citation at line 395. Line 167: move the reference to SODA to the line where you first talk about SODA, i.e. line 91; Thanks, we have added the citation at line 395. Line 179: The reference for Barnier 1995 is missing. In the reference there is a Barnier 2006. Please check. Thanks, we have added the reference for Barnier et al., 1995 at line 485. Line 185: the reference for Wang et al 2012 is missing. Thanks, we have added the reference for Wang et al., 2012 at line 638 Please check Line 201: move the reference to the Arakawa C-grid to where you first introduce the Arakawa C-grid, i.e. line 152. We have moved it to line 164 Line 228 : the reference WOA2005 is missing We have added the references Antonov et al., 2006 and Locarnini et al., 2006 3.3 Figures In general increase the size of the x/y labels especially in the maps and use higher resolution files so that the image does not loose sharpness when zooming in. Fig 1: change the colours for the cruises paths and the mooring station because now they cannot be easily distinguished from

the background Moreover, please indicate in Fig.1 more of the channels and seas you name in lines 30-33 to facilitate the nonexpert readers in understanding the unique geographical features of the SCS. Reduce the width (or the size in general to keep the aspect ratio) because the label of the scale is outside the printing area so it is missing in a printed version of the paper Thanks. We have tried to change colours for the cruises paths and the mooring station to the yellow, red and others, found the pink, green and black are the best colors to distinguish from the background. We also have added the names of channels and seas, such as Karimata Strait, Balabac Strait, Mindoro Strait, Java Sea, Sulu Sea, and East China Sea, and reduced the width. Fig.2: report in a separate plot the mean maximum and minimum AGV because it is currently difficult to compare them from the maps shown in Fig.2 or report these values explicitly in the text. Increase the sizes of the labels on the legend and axes. Use higher resolution files. The unity of measure is missing from the scale Also SSH bias maps can be added (MO minus AVISO and SCSOFS minus AVISO) to evidence better the behaviour of the two models with respect to the observations (analogous to Fig.4 maps) Thanks. We have changed color shaded of all plots from Sea Surface Height to the speeds of AVG in Fig. 2. Since we have not mentioned Sea Surface Height in the text. And we also have increased the sizes of the labels on the legend and axes and added the unity of measure of the scale. Maybe you can also change the color map for Figure 4: a red/white/blue (RWB) map is usually more appropriate to represent bias. For example in the actual maps the green color can correspond to bias values of both +0.5 and -0.5; using a RWB map would make more clear the areas where the bias is positive and where it is negative. Thanks, we have changed it follow your suggestion. Fig.5-6-7-8: units missing from the colorbar Thanks, we have added it. Fig.9: the correlation plots for salinity show a less good linear relationship between MO and SCSOFS and data with respect to the same plots for temperature? Did you try to plot the correlations in different depth ranges to see if the plots show a better linear correlation and if there is a depth range for which the correlation is not good and this degrades the overall linear relationship? Yes, we have tried to plot it in different depth, and got almost the

same results. So we just show the whole linear relationship in this paper. It is actually that the correlation for salinity is less than those for temperature. Fig 10: change the colormap so that also the pale yellow structures can be distinguished better from the white background Thanks, we have changed it follow your suggestion. Fig.11: explain what are the white areas on the map The white areas mean the SST increasing.

3.4 Language and typos Line 12-13: change "Mercator Ocean...in China" in "the global Mercator Ocean Operational System , developed and maintained by Mercator Ocean in France and the regional South China Sea Operational Forecasting System (SCSOFS) by the National Marine Environmental Forecasting Center (NMEFC) in China". I think it is better to underline that MO is a global ocean forecasting system developed by Mercator Ocean, a scientific institution in France. Line 22: change "AVISO data" in "satellite observations": at this point it is not yet clear to a medium reader what are AVISO data; change "results compared in above" in "outcome of the results compari-son" Line 42-43: change "in the NSCS....in the SSCS" in "is present in the NSCS, while a semiannual/biennial change from a cyclonic gyre regime in winter to an anti-cyclonic gyre regime in summer can be observed in the SSCS" Please note that the word "bian-nual" is ambiguous; some use it with the meaning of "twice per year" in which case it is better to use the word "semiannual", others use it to say "every two years", in which case it is better to say "biennial". Line 136 change "abnormal" in "outlier" Line 142/263: change "See" in "see" Line 145-147: change "All the measured...correlation analysis" in "The TS data collected from all the 5 cruises will be used to perform a correlation analysis of each of the simulated predictions of MO and SCSOFS models with the ob-servations." Line 149: change "Ocean" in "Oceanic" and modelling in "modeling". The first correction comes from the original name of ROMS while the second is to align the spelling of the word to the American English rule that you are using in other words (for example as in "analyzed") Line 208: the value "-1x1010 mËĘ4sËĘ-1" seems odd, please check in case there is a typo. Line 216: remove the comma before the paren-thesis Line 218 add a "-" between along and track Line 221: since it is the first time you mention SSH add the full name: Sea Surface Height (SSH) and remove the full name from line 232 Line 267 there must be a typo in "the range of large is less than". Please rewrite. Change "leading" in "anticipating" Line 317 change "ship" in "analysis" Line 320 remove "of relativity" Line 354 substitute "hot" with important Line 367 substitute "SST deceasing" with "SST decrease" Line 415 there is a typo: the first SCSOFS should be changed in MO

Thanks, we have changed all the typos in above in the revised paper.

Please also note the supplement to this comment:
http://www.nat-hazards-earth-syst-sci-discuss.net/nhess-2016-60/nhess-2016-60-AC1-supplement.pdf

**Supplement:**

L. Lusito (Referee)

letizia.lusito@cmcc.it

1. General comments

First of all, I would like to congratulate all the authors for the good scientific level of thework presented in this paper.This paper addresses the relevant problem of comparison and validation with real observations of two ocean models (SCSOFS and MO) in the South China Sea, an area which is becoming more and more strategic. Ocean models, in general, are the key to improve our knowledge of the sea state, present and future, on which to base political, environmental, economical decisions by governments and other stakeholders. In this respect, this issues are relevant not only to the general themes addressed by this journal, but also they are especially relevant in the context of this special issue subject 'Situational sea awareness technologies for maritime safety and marine environment protection'.

The results described here can be considered innovative; in fact, even if the South China Sea Ocean Forecasting System for the regional modelling of the south China Sea, and the global Ocean model, developed by Mercator Ocean, have been described in previous works, their comparison and validation with real observations in the South China Sea is a new important result that will allow a substantial improvements of the ocean forecasting capability in the area. Moreover the paper shows how both models could be improved regarding the forecast skills in case of devastating events like typhoons which are unfortunately not rare in that region.

The title is clear and reflects the content of the paper. The abstract provides a good summary of the results and of the conclusions that derive from them in a language easily understandable by the average reader. Every section describes clearly a particular aspect of the data, model or method used. The overall length is adequate and a good compromise between a too-long discussion containing a full description of many technical details and a too short description of the results without a proper introduction of the context, methods and data used.

It is strongly recommended that the English language is revised and improved by a native speaker although the actual form is sufficiently understandable. Some suggestions to improve the text will be reported separately.

Other comments (about units, figures, references, further clarifications) are reported in the next sections.

*We sincerely appreciate the reviewer for the constructive comments and suggestions. We addressed these concerns in the revision by following all the suggestions to significantly improve the manuscript in the following manner:*

1) *We have explained relative words or sentences and answered all the questions of the reviewer.*
2) *We have clarified the introductions to the satellite data, SLA, MGDSST, AVHRR in section 2.1.*
3) *We redraw or modified relative figures mentioned by the review, such as Fig. 1,2,4,5,6,7,8,10.*
4) *We revised all the styles, language, typos, references*

**2. Specific scientific comments**

**Line 17: explain or define better what you are referring to with the term "mesoscale activities"**

*We have explained it by adding the words ", such as ocean fronts, Typhoon, and mesoscale eddy,"*

**Line 74: explain better what you mean with the phrase "where they can then impact budgets of the tracers"**

*We have changed it to "where they can then impact budgets of heat, mass, momentum and biogeochemical properties".*

**Line 107: if the satellite data you use are from 2012, what is "April 2014" referring to? The starting date of the SSALTO/DUACS new data system?**

*The satellite datasets, DUACS 2014 version, are distributed to users in April 2014, which including the data from 1993 to present. We only use the data of 2012 to compare and validate our systems' results in this paper.*

**Line 109: quoting from the user manual of the L4 altimeter data you are using (and that you cite http://www.aviso.altimetry.fr/fileadmin/documents/data/tools/hdbk_duacs.pdf) "Change of resolution: in DUACS 2014 version, after the feedback from users, the Mercator grid projection with 1/3 x1/3 spatial resolution (Global product) is abandoned. The DUACS 2014 Global products are directly computed on a Cartesian 1/4 x1/4 spatial resolution." Therefore, depending which data you are using, it may be incorrect saying that the products are sampled from the Mercator gridded data so please, specify better if you are using the DUACS 2014 data or a former version (and which one). From the following discussion it seems you are using the DUACS 2014 data.**

*Yes, we use the DUACS 2014 data in this paper and have modified it in the revised manuscript.*

**Line 115: quoting the JMA database description of the MGDSST (http://neargoos1. jodc.go.jp/rdmdb/format/JMA/mgdsst.txt): "Merged satellite and in-situ data Global Daily Sea Surface Temperature (MGDSST): The MGDSSTs are analysed at the Office of Marine Prediction of the JMA with 1/4-degree grid resolution on the near realtime basis. SSTs derived from satellite's infrared sensors (AVHRR/NOAA) and microwave sensor (AMSR-E/AQUA), and in-situ SST (buoy and ship) are used in the analysis." The list of satellite products here does not match the list of satellite products you mention in the paper.**

*Thanks, we have revised it to match the description of the MGDSST in the website.*

**Line 122-127 please clarify in the paper that the reason for which you are using the "AVHRR-only" data is because the production of the AVHRR+AMSR data ended in 2011. Otherwise questions might arise on the impact of the usage of "AVHRR+AMSR" data in your analysis and how the results might differ from the results using "AVHRRonly" data.**

*Thanks, we have clarified it in the revised manuscript by adding the words "Since the production of the AVHRR+AMSR data ended in 2011" at line 145.*

**Line 137-138: just for my education: why to filter out the tidal signal you use a period of 25 hours and**

**not, for example 24 hours? Same, why did you use a 25-hour period to calculate the daily average?**

*It is because the periods of some of the major diurnal tidal constituents are longer than 24 hours. Actually 26 hours is better than 25 hours to filter out the tidal information.*

**Line 140 and following: you mention that there were 5 cruises to measure the temperature/salinity data but you use data from only two of them for the TS distribution comparison? Why only two? Why do you use the data from these particular cruises? Are the data from the remaining three cruises significantly different in some ways? What changes/improvements/impacts do you expect when using all the data available?**

*We did have compared both two systems with other three cruises, the results are almost the same. In order to save space, we only show two of them in this paper. The Qiongdong cruise is conducted in the coastal area, the Nansha cruise is conducted in the deep area. We also compared all the measured TS data from 5 cruises with the two system, please see Fig. 9 and section 3.2.3 Correlation ship between model and in-situ.*

**Line 244: be more specific than "little stronger than AVISO" For example you can add a time-series-like plot showing the basin-averaged, minimum and maximum velocities (in separated components u and v) for the two models and the satellite observations in the four months. On the x-axis you have the 4 time steps (Jan, Apr, Jul, Oct.), on the y-axis 3 u-velocities (min, max, basin-averaged) for each of the two models and the observation. Same for v. Alternatively, at least quote the basin-averaged velocities in the text. From the Figure only, it is very difficult to distinguish the length of the vectors hence to have an estimation of the magnitude of the velocities. Moreover, while on lines 243 you say that AVISO shows currents that are smaller than MO or SCSOFS, at lines 249 you say that AVISO has comparable velocities to MO and SCSOFS has smaller velocities so there is an incongruence in the text.**

*Thanks. It is not correct, we have deleted the words "except that the current speeds are a little stronger than AVISO".*

**Line 260: explain why in your opinion in spring the two models and the observation show each a different type of Kuroshio intrusion. Is this maybe due to some physics effects modelled differently in the two models or boundary conditions not implemented in the best way in the two models. Can you also explain why this effect is visible mainly in spring?**

*It is obviously the types of Kuroshio intrusion are different from each among the three results, you can refer the paper Nan et al. (2011a). And AVISO, MO, and SCSOFS show the leaking path, looping path, and leaping path in spring, respectively. Actually, I have not researched much on this problem. I think it is an interesting scientific problem, and additional work need to do to study it in detail. But this is out of the scope of this paper.*

*In my opinion, it may due to the surface wind forcing are different for the two models. Since the wind is weaker in spring than other seasons in this area, the wind forcing used in both two models may not agreement well with the real wind.*

**Line 271: I would prefer to use the word "temporal" instead of "phase" bias in this case. The word phase is generally more used when speaking about angles. Do you have an explanation about this temporal behaviour of SCSOFS?**

*Thanks, we have changed the word from "phase" to "temporal". It may due to the surface wind forcing, we will double check about it.*

**At line 281 you say that the SST has been assimilated in SCSOFS but this is not mentioned when you describe which data are assimilated in the model in lines 187 and 188.**

*Yes, the SST data has been assimilated in SCSOFS by the thermal feedback method to correct net surface*

*heat flux (Barnier et al., 1995). We have modified the description at Line 189 and 190.*

**At line 286 you say that the SST variation is larger in winter and smaller in summer for both MO and SCSOFS but when looking at Figure 4 this appears to not be true: the absolute magnitude of the variation is much larger for SCSOFS (where you see large blue and red areas) than for MO (where you see prevalent green everywhere) and has the same values (but with opposite signs) for summer and winter; in winter the area that show a large variation is simply larger so it is better to specify the basin-averaged variation is larger. What it seems more correct is that SCSOFS overestimates the SST in winter and underestimates the SST in summer, therefore has a more uniform trend of the SST through the year with respect to MO**

*Thanks, we have modified it at line 286.*

**Line 287 you quote 3 values for RMSE for each of the models saying that they are the maximum minimum mean monthly but it is no clear what do you mean with "monthly": are these the averaged values over the 4 months or do they correspond to a specific month? In any case please quote the values for each month for a better comparison of the performances of MO and SCSOFS**

*Thanks, we have modified it at line 298 in the revised version.*

**Line 303: you say that the isohaline is located at 50 km for in-situ data and SCSOFS and at 20 km for MO but this big difference is not so evident in Figure 6, therefore from Figure 6 it cannot be concluded that SCSOFS performs better than MO when compared to in-situ data. A plot of a vertical profile of TS and the TS bias for let's say 20km, 50 km and 70 km can clarify better the performances of the models in this case.**

*We have changed the plot of SCSOFS in Figure 6, and revised the position of MO from 20km to 40km.*

**Line 375 Define what is W**

*W is the Okubo-Weiss parameter (Xiu et al., 2010) defined as:*

$$W = S_n^2 + S_s^2 - \omega^2,$$

*Where*

$$S_n = \frac{\partial u}{\partial x} - \frac{\partial v}{\partial y}$$

$$S_s = \frac{\partial v}{\partial x} + \frac{\partial u}{\partial y}$$

$$\omega = \frac{\partial v}{\partial x} - \frac{\partial u}{\partial y}$$

**Line 387 Why do you use a period of 24 years for SCSOFS and just one year for MO?**

*Since we have got only the year of 2012 data from MO, only one year of MO data is available in this study, and 14 years, from 2001 to 2014 for SCSOFS.*

**Line 391 You say that all the three results show a small number of eddies in autumn and a larger number of eddies in spring but this is not true for MO when looking at table 1, where MO predicts 13 total eddies for both spring and autumn**

*Thanks, we have revised it as:" As for the seasonal distributions (figures not shown), all three data have most eddies in spring. Both AVISO and SCSOFS have lest eddies in fall and more cyclones than anti-cyclones in spring and fall, and all three have less cyclones than anti-cyclones in summer."*

**Line 395 explain better the oversimplification of SLA calculation for MO and why SCSOFS does not suffer from this.**

*Oversimplification hear means SLA calculated by only one year's MO data averaged and extracted from*

*SSH. This may introduce great error for SLA and the eddy identification.*

**3. Style comments and suggestions**

**3.1 General**

**Be consistent with the space between the value and the unit of measure, for example in line 28 you write "1200m" and line 156 you write "0.16 m". As a general reference, the NIST Guide for the Use of the International System of Units (SI) states "7.2 Space between numerical value and unit symbol: In the expression for the value of a quantity, the unit symbol is placed after the numerical value and a space is left between the numerical value and the unit symbol. The only exceptions to this rule are for the unit symbols for degree, minute, and second for plane angle (...) in which case no space is**

**left between the numerical value and the unit symbol."**

*Thanks, we have modified all the expressions in the revised paper.*

**Change all the "northeastly" in "northeasterly" , "southwestly" in "southwesterly" and all the similar words.**

*Thanks, we have changed them in the revised paper.*

**Use consistently "coastal currents" or "Coastal Currents" (check for examples line 50-51)**

*Thanks, we have changed them in the revised paper.*

**Be consistent in using 1/4 or 0.25 for the horizontal resolutions (for example lines 116 and 119)**

*Thanks, we have changed them in the revised paper.*

**In the section about MO (2.4), you keep calling the model PSY4V1R3 systematically without any mention of Mercator Ocean and in the later sections this name (PSY4V1R3) disappears. Please introduce at some point in section 2.4 a clarification like : the PSY4V1R3 configuration described here is indicated for as MO model through this paper.**

*Thanks, we have clarified it in the revised paper.*

**3.2 References**

**Bell, 2015 is never used; Chu, 2001 is never used; Daudin, 2013 never used;**

*Thanks, we have deleted these references.*

**Weiss 1991, not used;**

*Thanks, we have added the citation at line 395.*

**Line 167: move the reference to SODA to the line where you first talk about SODA, i.e. line 91;**

*Thanks, we have added the citation at line 395.*

**Line 179: The reference for Barnier 1995 is missing. In the reference there is a Barnier 2006. Please check.**

*Thanks, we have added the reference for Barnier et al., 1995 at line 485.*

**Line 185: the reference for Wang et al 2012 is missing.**

*Thanks, we have added the reference for Wang et al., 2012 at line 638*

**Please check Line 201: move the reference to the Arakawa C-grid to where you first introduce the Arakawa C-grid, i.e. line 152.**

*We have moved it to line 164*

**Line 228 : the reference WOA2005 is missing**

*We have added the references Antonov et al., 2006 and Locarnini et al., 2006*

**3.3 Figures**

**In general increase the size of the x/y labels especially in the maps and use higher resolution files so that the image does not loose sharpness when zooming in.**

**Fig 1: change the colours for the cruises paths and the mooring station because now they cannot be**

easily distinguished from the background Moreover, please indicate in Fig.1 more of the channels and seas you name in lines 30-33 to facilitate the nonexpert readers in understanding the unique geographical features of the SCS. Reduce the width (or the size in general to keep the aspect ratio) because the label of the scale is outside the printing area so it is missing in a printed version of the paper

*Thanks. We have tried to change colours for the cruises paths and the mooring station to the yellow, red and others, found the pink, green and black are the best colors to distinguish from the background. We also have added the names of channels and seas, such as Karimata Strait, Balabac Strait, Mindoro Strait, Java Sea, Sulu Sea, and East China Sea, and reduced the width.*

Fig.2: report in a separate plot the mean maximum and minimum AGV because it is currently difficult to compare them from the maps shown in Fig.2 or report these values explicitly in the text. Increase the sizes of the labels on the legend and axes. Use higher resolution files. The unity of measure is missing from the scale Also SSH bias maps can be added (MO minus AVISO and SCSOFS minus AVISO) to evidence better the behaviour of the two models with respect to the observations (analogous to Fig.4 maps)

*Thanks. We have changed color shaded of all plots from Sea Surface Height to the speeds of AVG in Fig. 2. Since we have not mentioned Sea Surface Height in the text. And we also have increased the sizes of the labels on the legend and axes and added the unity of measure of the scale.*

Maybe you can also change the color map for Figure 4: a red/white/blue (RWB) map is usually more appropriate to represent bias. For example in the actual maps the green color can correspond to bias values of both +0.5 and -0.5; using a RWB map would make more clear the areas where the bias is positive and where it is negative.

*Thanks, we have changed it follow your suggestion.*

Fig.5-6-7-8: units missing from the colorbar

*Thanks, we have added it.*

Fig.9: the correlation plots for salinity show a less good linear relationship between MO and SCSOFS and data with respect to the same plots for temperature? Did you try to plot the correlations in different depth ranges to see if the plots show a better linear correlation and if there is a depth range for which the correlation is not good and this degrades the overall linear relationship?

*Yes, we have tried to plot it in different depth, and got almost the same results. So we just show the whole linear relationship in this paper. It is actually that the correlation for salinity is less than those for temperature.*

Fig 10: change the colormap so that also the pale yellow structures can be distinguished better from the white background

*Thanks, we have changed it follow your suggestion.*

Fig.11: explain what are the white areas on the map

*The white areas mean the SST increasing.*

**3.4 Language and typos**

Line 12-13: change "Mercator Ocean...in China" in "the global Mercator Ocean Operational System , developed and maintained by Mercator Ocean in France and the regional South China Sea Operational Forecasting System (SCSOFS) by the National Marine Environmental Forecasting Center (NMEFC) in China". I think it is better to underline that MO is a global ocean forecasting system developed by Mercator Ocean, a scientific institution in France.

**Line 22:** change "AVISO data" in "satellite observations": at this point it is not yet clear to a medium reader what are AVISO data; change "results compared in above" in "outcome of the results comparison"

**Line 42-43:** change "in the NSCS....in the SSCS" in "is present in the NSCS, while a semiannual/biennial change from a cyclonic gyre regime in winter to an anti-cyclonic gyre regime in summer can be observed in the SSCS" Please note that the word "biannual" is ambiguous; some use it with the meaning of "twice per year" in which case it is better to use the word "semiannual", others use it to say "every two years", in which case it is better to say "biennial".

**Line 136** change "abnormal" in "outlier"

**Line 142/263:** change "See" in "see"

**Line 145-147:** change "All the measured...correlation analysis" in "The TS data collected from all the 5 cruises will be used to perform a correlation analysis of each of the simulated predictions of MO and SCSOFS models with the observations."

**Line 149:** change "Ocean" in "Oceanic" and modelling in "modeling". The first correction comes from the original name of ROMS while the second is to align the spelling of the word to the American English rule that you are using in other words (for example as in "analyzed")

**Line 208:** the value "-1x1010 m^4s^-1" seems odd, please check in case there is a typo.

**Line 216:** remove the comma before the parenthesis

**Line 218** add a "-" between along and track

**Line 221:** since it is the first time you mention SSH add the full name: Sea Surface Height (SSH) and remove the full name from line 232

**Line 267** there must be a typo in "the range of large is less than". Please rewrite. Change "leading" in "anticipating"

**Line 317** change "ship" in "analysis"

**Line 320** remove "of relativity"

**Line 354** substitute "hot" with important

**Line 367** substitute "SST deceasing" with "SST decrease"

**Line 415** there is a typo: the first SCSOFS should be changed in MO

*Thanks, we have changed all the typos in above in the revised paper.*

**Interactive comment on Nat. Hazards Earth Syst. Sci. Discuss., doi:10.5194/nhess-2016-60, 2016.**

**Comparison and validation of global and regional ocean forecasting systems in the South China Sea**

Xueming Zhu[1], Hui Wang[1], Guimei Liu[1*], Charly Régnier[2], Xiaodi Kuang[1], Dakui Wang[1], Shihe Ren[1], Zhiyou Jing[3], Marie Drévillon[2]

[1]Key Laboratory of Research on Marine Hazards Forecasting, National Marine Environmental Forecasting Center, Beijing, 100081, China

[2]Mercator Océan, Ramonville Saint Agne, France

[3]State Key Laboratory of Tropical Oceanography, South China Sea Institute of Oceanology, Chinese Academy of Sciences, Guangzhou, 510301, China

*Correspondence to*: Guimei Liu (liugm@nmefc.gov.cn)

**Abstract.** In this paper, the performances of two operational ocean forecasting systems, the global Mercator Océan (MO) Operational System, developed and maintained by Mercator Océan in France and the regional South China Sea Operational Forecasting System (SCSOFS) by the National Marine Environmental Forecasting Center (NMEFC) in China, have been examined. Both systems can provide science-based nowcast/forecast products, such as temperature, salinity, water level and ocean circulations. Based on the observed satellite and *in-situ* data have been obtained in 2012 in the South China Sea, the comparison and validation of the ocean circulations, the structures of the temperature and salinity, and some mesoscale activities, such as ocean fronts, Typhoon, and mesoscale eddy, are shown. Comparing with the observation, the ocean circulations and SST of MO show better results than those of SCSOFS. However, the structures of temperature and salinity of SCSOFS are better than those of MO. For the mesoscale activities, SST fronts and SST decreasing during the typhoon Tembin of SCSOFS are better agreement with the previous study or satellite data than those of MO; but both of them show some differences from satellite observations. Finally, according to the outcome of the results comparison, some suggestions have been proposed for both systems to improve their performances in the near further.

**Keywords.** SCSOFS, Mercator Océan, South China Sea, Operational Forecasting System

**1 Introduction**

The South China Sea (SCS, Fig.1) is the largest and deepest semi-enclosed marginal sea of the Northwestern Pacific (NWP), with the area is about 3.5 million $km^2$, the mean and maximum depth is about 1200 m and 5300 m, respectively. A wide continental shelf with depth less than 200 m located in the northern SCS (NSCS). There are numerous islands, reefs, beaches, shoals in large basin of the southern SCS (SSCS). It is connected with the adjacent seas through a number of channels, to the East China Sea in north, the NWP in east, the Sulu Sea in southeast, and the Java Sea in south, by the Taiwan Strait (TWS), the Luzon Strait (LUS), the Mindoro Strait and the Balabac Strait, the Karimata Strait, respectively. Its unique geographical features, rich marine mineral and petroleum resources play a significant role to many countries adjacent to it.

The SCS is located in the East Asian Monsoon (EAM) winds regime, the northeasterly winds usually prevail with an average wind speed of 9 m/s over the whole domain in winter, while the southwesterly winds prevail with an average magnitude of 6m/s dominating over the most parts of the SCS in summer (Hellerman and Rosenstein, 1983). The EAM is considered to be the main factor for driving the upper layer basin-scale circulation pattern in the entire SCS, showing an obvious seasonal variation with a cyclonic gyre in winter and an anti-cyclonic gyre in summer (Wyrtki, 1961; Mao et al., 1999; Wu et al.,

1999; Qu, 2000; Chu and Li, 2000). However, some other literatures insist that a persistent cyclonic gyre is present in the NSCS, while a semiannual change from a cyclonic gyre in winter to an anti-cyclonic gyre regime in summer can be observed 
[revised manuscript text omitted]

---

## Referee Comment (RC2) · Anonymous Referee #2 · 28 Apr 2016

This paper assessed and validated the performance of two operational ocean forecasting systems, the global Mercator Ocean operational forecast system developed in France and the regional South China Sea operational forecasting system developed in the National Marine Environmental Forecasting Center in China by comparing the model results from the two systems with a comprehensive set of observations from satellite and in-situ measurements. The recommendations were proposed for future improvement of both systems based on comparison results. The methodology and matrix of evaluation and validation are reasonable, the observations are well quality-controlled. My detail edits were added in the manuscript with review "track changes" and sent the revised Microsoft word document back to the authors. I would recommend accept this paper for publication after further editing of grammars.

---

## Editor Comment (EC1) · I Federico (Editor) · 3 May 2016

**Comparison and validation of global and regional ocean forecasting systems for the South China Sea**

Xueming Zhu[1], Hui Wang[1], Guimei Liu[1*], Charly Régnier[2], Xiaodi Kuang[1], Dakui Wang[1], Shihe Ren[1], Zhiyou Jing[3], Marie Drévillon[2]

[1]Key Laboratory of Research on Marine Hazards Forecasting, National Marine Environmental Forecasting Center, Beijing, 100081, China
[2]Mercator Océan, Ramonville Saint Agne, France
[3]State Key Laboratory of Tropical Oceanography, South China Sea Institute of Oceanology, Chinese Academy of Sciences, Guangzhou, 510301, China

*Correspondence to*: Guimei Liu (liugm@nmefc.gov.cn)

**Abstract.** In this paper, the performance of two operational ocean forecasting systems, the global

Mercator Océan (MO) Operational System, developed and maintained by Mercator Océan in France and the regional South China Sea Operational Forecasting System (SCSOFS) by the National Marine

Environmental Forecasting Center (NMEFC) in China, have been examined. Both systems can provide science-based nowcast/forecast products of temperature, salinity, water level and ocean circulations.

comparison and validation of the ocean circulations, the structures of the temperature and salinity, and some mesoscale activities, such as ocean fronts, Typhoon, and mesoscale eddy, are cpnducted  based on the observed satellite and *in-situ* data obtained in 2012 in the South China

Sea. The results showed that MO performs better in forecasting the ocean circulations and SST , and SCSOFS

performs better in simulating the structures of temperature and salinity

. For the mesoscale activities, SCSOFS performance is better than MO in simulating SST fronts and SST decreasing during the typhoon Tembin  compared with the previous study or satellite data ; but model results from both of SCSOFS and

MO show some differences from satellite observations.

 In conclusion, some recommendations have been proposed for both forecast systems to improve their forecasting performance in the near further based on our comparison and validation.

**Keywords.** SCSOFS, Mercator Océan, South China Sea, Operational Forecasting System

**1 Introduction**

The South China Sea (SCS, Fig.1) is the largest and deepest semi-enclosed marginal sea of the

Northwestern Pacific (NWP), with the area  about 3.5 million km$^2$, the mean and maximum depth ranging from 1200 m and 5300 m, respectively. The northern SCS (NSCS) is a wide continental shelf with depth less than 200 m , and the southern SCS (SSCS)

comprises numerous islands, reefs, beaches, shoals in large basin .

SCS is connected with the adjacent seas through a number of channels, to the East China Sea in north, to the Northwest Pacific Ocean (NWP) in east, to the Sulu Sea in southeast, and to the Java Sea in south, by the Taiwan Strait (TWS), the Luzon Strait (LUS), the Mindoro Strait and the Balabac Strait, the

Karimata Strait, respectively. SCS has its unique geographical features, rich marine mineral and petroleum resources so that it is very important for many countries adjacent to it.

The SCS is located in the East Asian Monsoon (EAM) winds regime, the northeasterly winds usually prevail with an average wind speed of 9 m/s over the whole domain in winter, while the southwesterly winds prevail with an average magnitude of 6 m/s dominating over the most parts of the SCS in summer (Hellerman and Rosenstein, 1983). The EAM is considered to be the major factors for driving the upper layer basin-scale circulation pattern in the entire SCS, showing an obvious seasonal variation with a cyclonic gyre in winter and an anti-cyclonic gyre in summer (Wyrtki, 1961; Mao et al., 1999; Wu et al.,

1999; Qu, 2000; Chu and Li, 2000). However, some other literatures insist that a persistent cyclonic gyre is present in the NSCS, while a semiannual changing from a cyclonic gyre in winter to an anti-cyclonic gyre regime in summer can be observed 
[revised manuscript text omitted]
* SST (buoy and ship) data Global Daily SST (MGDSST), with a 0.25°×0.25° horizontal resolution, which are analysed and published at the Office of Marine Prediction of the Japan Meteorological Agency (JMA). The data can be obtained from http://near-goos1.jodc.go.jp/.

The other one is derived from the NOAA 0.25°×0.25° daily Optimum Interpolation Sea Surface

Temperature (OISST), which is an analysis constructed by combining observations from different platforms, such as satellites, ships, buoys, on a regular grid via optimum interpolation. Right now,

National Centers for Environmental Information (NCEI) provides two kinds of OISST: one uses infrared satellite data from the Advanced Very High Resolution Radiometer (AVHRR) named as AVHRR-only, and the other one uses AVHRR data along with microwave data from the Advanced Microwave

Scanning Radiometer (AMSR) on the Earth Observing System Aqua or AMSR-E satellite named as

AVHRR+AMSR. Since the production of the AVHRR+AMSR data ended in 2011, the first one,

AVHRR-only, is used in this study, which spans 1981 to the present and can be downloaded from the website http://www.ncdc.noaa.gov/oisst/data-access.

**2.2 *In-situ* data**

The *in-situ* data employed in this paper for the comparison and validation of both systems are provided by the South China Sea Institute of Oceanology, Chinese Academy of Sciences. There were one mooring to measure the sea water velocity and 5 cruises conducted to measure the temperature and salinity (TS) in the SCS during 2012.

The mooring station is located at Maoming (Fig. 1), where bottom-mounted upward-looking 75 kHz

Acoustic Doppler Current Profilers (ADCPs) were deployed to monitor the current profile (U

component and V component) from the depth of 2 m to 48 m with a 2-m interval in vertical. The period of the monitoring is from 11 July to 8 October, in 2012, with a temporal interval 10 min. Firstly, the outlier data are eliminated from the original measured data;  second, a low-pass filter with 25-hour is applied to remove tidal current; and daily mean currents are calculated using a 25-hour average , which were used  to compare and validate with the simulated results of MO and SCSOFS.

The TS data from the five cruises were measured by SeaBird 19 plus conductivity-temperature-depth (CTD) with 1-m resolution in vertical. Among the five cruises, one is the Qiongdong cruise in the NSCS, which was conducted for 9 days from 12 to 20 July at 90 stations along 6 sections (sSee Fig.1); another one is the Nansha cruise around the Nansha Islands, which was conducted for 5 days from 24 to 28 August at 17 stations along 10°N section from 109.5°E to 117.5°E. The TS data from those two cruises will be used to compare and validate the TS distribution from MO and SCSOFS in vertical and horizontal. All the measured The TS data collected from all the 5 cruises will be usedcollected to perform a correlation analysis of each of the simulated predictions of compare with the simulated data from MO and SCSOFS models with the obervationsvia correlation analysis.

**2.3 The configurations of SCSOFS**

The SCSOFS usesis build up based on the Regional Oceanic Modelling System (ROMS), which is a three-dimensional, non-linear primitive equations, free surface, hydrostatic, split-explicit, topography-following-coordinate in vertical and orthogonal curvilinear in horizontal on a staggered Arakawa C-grid (Arakawa and Lamb, 1977) oceanic model (Shchepetkin and McWilliams, 2005).

To avoid the influences of boundary to the circulations in the SCS, the model's boundaries was extended to southward and eastward, then the model covered a larger domain (4.5°S to 28.3°N, 99°E to 145°E, Fig. 1) than the SCS. The horizontal resolution variates from 1/12° in the south and east boundary to 1/30° in the SCS. There were 36 s-coordinate levels in the vertical with the thinnest layer being 0.16 m on the surface. The bathymetry was extracted from the ETOPO1 data sets published by U.S. National Geophysical Data Center (NGDC), which is a global relief model of Earth's surface that integrates ocean bathymetry and land topography, with 1 arc-minute resolution (Amante and Eakins, 2009). The ETOPO1 dataset has combined the satellite altimeter observations, shipping load sonar measurement, multi resolutions digital terrain database and the global digital terrain model and many other data sources, and it has been used in the global and regional oceanic models widely. And the original ETOPO1 bathymetry was revised in the area of nearnext to the coast of China mainland according to the *in-situ* data collected in NMEFCmeasured by our group, then smoothed according to Shapiro (1975). The maximum depth was set to be 6000 m and the minimum depth to be 10 m in the model (Wang, 1996).

The initial temperature and salinity conditions were derived from the climatology monthly mean Simple Ocean Data Assimilation (SODA, Carton and Giese, 2008) in January. However, the initial velocities and elevation were set to zero, which means to integrate the model from a static status. The model's western lateral boundary was treated as a wall. The other three (northern, southern, eastern) lateral boundaries were opened, whose temperature, salinity, velocity, and elevation were prescribed by spatial interpolation of the monthly mean SODA dataset. The 2D and 3D velocities, through the open boundaries, are modulated to guarantee the conservation of volume flux in the whole model domain. In addition, the nudging technology was used for 3D velocity, temperature, and salinity to the three open lateral boundaries with a 30-day time scale for outflow and 3-day for inflow.

The model is forced using 6-hourly wind stress, net fresh water flux, net heat flux, surface solar shortwave radiation at surface from NCEP_Reanalysis 2 data provided by the NOAA/OAR/ESRL PSD,

Boulder, Colorado, USA, accessible from the web site at http://www.esrl.noaa.gov/psd/ (Kanamitsu et al., 2002). In order to get more reasonable simulated SST, the kinematic surface net heat flux sensitivity to SST (dQ/dSST) is used to introduce thermal feedback to correct net surface heat flux (Barnier et al.,

1995) with a constant number -30 $W/m^2/$ $^\circ C$ the whole domain. The MGDSST data is used to correct net surface heat flux. In addition, the monthly mean climatology discharges of the Mekong River and the

Pearl River are prescribed to the model.

The system was run with 6 seconds time step for the external mode, and 180 seconds for the internal mode under the initial conditions, boundary conditions and surface forcing mentioned in above. The system was conducted a hindcast run from 2000 to 2011 after a 15 years climatology run for spin-up (Wang et al., 2012). The model results were archived to the snapshot with a 5-day interval, which were used as the ensemble members for the EnOI (Ensemble Optimal Interpolation) method assimilation. After the hindcast run, the system was conducted an assimilation run in 2012 with EnOI

method, the along track SLA data from AVISO had been assimilated as the observations with a 7-day time window. The details on the EnOI applied in the SCSOFS can be referred as Ji et al. (2015). The assimilated results were archived to daily mean with a 1-day interval in 2012, which were used to compare and validate in this paper. Then the system is implemented into operations in NMEFC

since January 1st, 2013. It runs on daily bases for 6 day simulations (1-day nowcast and 5-day forecast)

 provide 120-hour forecasting products of the three dimensional  ocean temperature, salinity and currents with 24 hour interval.

**2.4 The configurations of MO**

The high resolution global analysis and forecasting system PSY4V1R3 was operational as the V2 of the

MyOcean project from February 2011 up to April 2013, when it was replaced by the PSY4V2R2 system.

During this period, PSY4V1R3 has been producing weekly 14-day hindcasts and daily 7-day forecasts.

**The PSY4V1R3 configuration described as followed is indicated for as MO model through this paper.**

The model configuration of PSY4V1R3 is based on a tripolar ORCA grid type (Madec and Imbard,

1996) in the NEMO 1.09 version with a 1/12° horizontal resolution which means 9 km at the Equator, 7

km at mid latitudes and 2km toward the Ross and Weddel Sea. The grid cells follow an Arakawa C-grid type (Arakawa and Lamb, 1977). The 50-level vertical discretization retained in this system has 1m resolution at the surface, decreasing to 450 m at the bottom and 22 levels within the upper 100 m. "Partial cell" parametrization was chosen for a better representation of the topographic floor (Barnier et al.,

2006). The high frequency gravity waves are filtered out by the free surface formulation of Roullet and

Madec (2000).

For the diffusion, a horizontal bilaplacian was added along the equator (20 $m^2s^{-1}$) and two laplacians in the Canadian straits (up to 100 $m^2s^{-1}$). Laplacian lateral isopycnal diffusion was added on tracers (125

$m^2s^{-1}$) and a horizontal biharmonic viscosity was added for the momentum ($-1\times1010$ $m^4s^{-1}$ at the

[revised manuscript text omitted]

**Comment [ZA1]:** What are three results referring? Results from MO and SCSOFS, what is the third? (geostrophic flow?)

**Comment [ZA2]:** See above comments by AVISO, MO, SCSOFS, respectively. The westward intensification along the eastern coast of the Vietnam is morest obvious in autumn than other three seasons, and the maximum speed is more than 1.0m s$^{-1}$ for MO and SCSOFS, but 0.7 m s$^{-1}$ for AVISO.

As mentioned in Sect. 1, the Kuroshio intruding the SCS through the LUS has been distinguished by three types as the looping path, the leaking path and the leaping path, according to Nan et al. (2011a). All three results show the looping path in winter, the leaping path in summer and leaking path in autumn, which is well consistentaccordance with the model results showed by Wu and Chiang (2007). However, AVISO, MO, and SCSOFS show the leaking path, looping path, and leaping path in spring, respectively.

**3.1.2 Time series from mooring station**

Figure 3 shows the comparison of the daily mean time series of the u, v components from the mooring, MO, and SCSOFS atin 40m-depth layer at the Maoming station (sSee Fig. 1) from July 11 to October 8, 2012. Both MO and SCSOFS can capturecatch the samilare variation trends of the time series with the mooring observation. Especially, MO results match have represented the observed current variations well for both u- and v- component, during the period of the Typhoon Kai-tak on 17 August 2012. Although SCSOFS shows the larger velocity during the Typhoon Kai-tak, the maximum velocity range of large is less than the observation and anticipating leading the observation about 1 day. The root mean square errors (RMSE) between observations and models of MO and SCSOFS and observation are 0.075 m s$^{-1}$, 0.094 m s$^{-1}$ for u-component, 0.062 m s$^{-1}$, 0.084m s$^{-1}$ for v-component, respectively. Overall, MO results are in better agreement with the observations than those of SCSOFS. However, SCSOFS results have a temporalphase bias (phase shift) comparing with the observation, which is leading the observations about 1 day.

**3.2 Temperature and Salinity**

**3.2.1 SST**

SST is a very important prognostic variable in a hydrostatic ocean general circulation numerical model, which plays a key role to the ocean circulations and the air-sea interaction. So SST error is a crucial criteria of the numerical model skill, especially for an operational ocean circulation model. In fact, the SST simulation error is affected by several factors, for example the limitation of physical model, the surface atmospherice forcing conditions, the bias of initial field and the uncertainty from the open boundary, as pointed out by Ji et al. (2015). Although the SST data have been assimilated into both MO

and SCSOFS, the assimilated SST still has some errors for both systems.

Figure 4 shows the distributions of the monthly mean SST errors between two systems and MGDSST in the SCS in 2012. The errors show an obvious regional distribution, the larger errors mainly appear in the coastal region for the depth shallower than 200 m, such as in the TWS, the eastern of the Guangdong province in January, the gulf of Tonkin in July.  The strong seasonal variation for the basin-averaged SST error can also  be found, which is larger in winter and smaller in summer fo both systems. Comparing with MGDSST, the maximum, minimum, and mean for the basin-averaged 12 monthly RMSEs are 0.78 ℃, 0.37 ℃, 0.51 ℃ for the MO, 1.15 ℃, 0.56 ℃, 0.86 ℃ for the SCSOFS, respectively, in the SCS. Based on the Fig. 4, MO performed better than SCSOFS in simulating SST  by comparing with

MGDSST.

**3.2.2 Horizontal and vertical distribution of TS**

Water temperature and salinity horizontal distributions at 10-m depth layer  in the eastern of

Hainan island from the *in-situ* observations of Qiongdong cruise, model results from MO, and SCSOFS

are shown in Fig. 5. Two clear cold and salty water cores located at the eastern of Hainan island, which are located at about (110.75°E, 19.2°N) and (111.3°E, 19.7°N), are shown in both

*in-situ* observations and SCSOFS (Fig. 5) with the cores from SCSOFS being more saline than the *in-situ* observations. It can be easily deduced that the two cores are produced by upwelling process from the TS vertical distributions of the section K, F, H, and G (Jing et al., 2015).

Figure 6 shows the vertical temperature and salinity distributions from the *in-situ* observations of

Qiongdong cruise, model results from MO and SCSOFS, along section E. Both systems have got the simila vertical structures of TS with the *in-situ* observations. All of them demostrated the obvious upwelling system, with cold and salty waters flowing from offshore to nearshore along the bottom. All three results show the upper mixing layer depth is about 15 m The sea water is well mixed above 15 m depth and the isotherms and isohalines are almost vertical, where  strong vertical stratification is shown in summer. The diluted water is flushing from the nearshore to offshore, with the 33-isohaline cross with the sea surface  at the position of about 50 km from the coast for both *in-situ* observations and SCSOFS, but at the position of about 420 km for MO. In above, it is indicated that the results of SCSOFS is better agreement with the *in-situ* observations than those of MO. The vertical distributions of TS from the *in-situ* observations of Nansha cruise, and model results from MO, and SCSOFS along the 10°N section are shown in Fig.7 for the layer above 300 m and Fig.8 for the layer of 300-1200 m. Both systems have got almost the same vertical structures with the *in-situ*, especially for the upper mixing layer depth about 70 m are shown in the three results. WaterThe temperature almost linearly decreases from 28 ℃ to 3℃ with the depth going deep increasing from the bottom of the upper mixing layer to the 1200 m depth. However, the salinity increases from 33.5 psu to 34.5 psu with the depth going deep increasing from the bottom of the upper mixing layer to about 200m depth, and keeps almost constant at 34.5 psu from 200 m to 300 m depth. Then a fresher water layer exists in the middle layer from about 400 m to 700 m with the salinity about 34.4 psu. Below the middle layer, the salinity again increases from 34.4 to 34.58 with the depth increasing from 700 m to 1200 m. It indicates that the results of MO and SCSOFS are in goodwell agreement with *in-situ* observations, except that the salinity of the fresh water in the middle layer from MO is less than 34.4 whichand is fresher than those of *in-situ* and SCSOFS, but the thickness of the fresh layer is thicker than those of *in-situ* and SCSOFS.

**3.2.3 Correlation analysisship between model and *in-situ***

In order to better compare and validate the performances of the two systems, we collected all the measured TS data from five cruises in the SCS in 2012 to conduct a comprehensive correlation analysis. Figure 9 shows the comparison of relativity of TS between MO, SCSOFS and *in-situ* by scatter points, respectively. Any point in the Fig.9 is corresponded with two values of temperature or salinity, one is from the *in-situ* along X axis, and the other one is from MO or SCSOFS along Y axis. The correlation coefficients of temperature are 0.987, 0.982, and of salinity are 0.717, 0.897, between MO, SCSOFS and *in-situ*, over the 95% significance level, respectively, which is showing the good relativity between MO, SCSOFS and *in-situ*. It also indicates that the relativity of temperature is in better agreement with *in-situ* than those of salinity for both MO and SCSOFS, and SCSOFS is in better agreement with *in-situ* than MO for salinity.

Comment [ZA3]: It is not clear which layer

**3.3 Mesoscale activities**

**3.3.1 SST front**

Oceanic front is a good indicator for connection between water masses with different hydrological features, which is an important marine mesoscale phenomenon. There are numerous SST fronts in the

SCS, most of them located on the continental shelf with the depth below 200 m or aligned with the shelf break, especially in the NSCS. A few obvious SST fronts have been identified from the long-term

NOAA/NASA Pathfinder SST data, namely: Fujian-Guangdong Coastal Front, Pear River Estuary

Coastal Front, Taiwan Bank Front, Kuroshio Intrusion Front, Hainan Island East Coastal Front, Tonkin

Gulf Coastal Front (Wang et al., 2001). All of them exhibit very strong seasonal variability, which is mainly due to the EAM (Belkin and Cornillon, 2003).

Figure 10 shows the distributions of SST fronts from MO and SCSOFS for four seasons. The similar frontal patterns with their evident seasonal variations are shown in both systems, except for some small differences. In winter, most fronts reach maximum strength (>0.2 $^{\circ}$C/km). The Fujian-Guangdong

Coastal Front and Taiwan Bank Front are major fronts in the SCS which agree with previous satellite result from Wang et al. (2001). These two fronts merge and extend to Pearl River Estuary and the

Hainan Island. The Hainan Island East Coastal Front is stronger in MO than in SCSOFS, whereas the

Tonkin Gulf Coastal Front is stronger in SCSOFS than in MO. The Kuroshio Intrusion front is obvious in SCSOFS, however, it is hardly seen in MO. In spring, most fronts become weak obviously due to the weakening of northeast monsoon fom both systems, except that the Hainan

West Coastal Front emerges in SCSOFS. In summer, weakening almost occurs in all the fronts mentioned above for SCSOFS, which is in agreement with the result of Wang et al. (2001). However, disappearing occurs in all the fronts for MO. In fall, most fronts fade and disappear, except that the

Taiwan Bank front has very weak strength compared to other seasons for both systems. Both systems have not shown the Kuroshio Intrusion Front identified by Wang et al. (2001) in summer and fall.

**3.3.2 The Typhoon Tembin**

There are a lot of typhoons in the SCS during the typhoon season in every year, so that the typhoon activities are very frequent in the SCS, especially in 2012. One important study on the air-sea interaction is the responding of the physical ocean dynamics to typhoon in the oceanic upper layer. One important responding is the decreasing of SST due to the strong vertical mixing caused by typhoon (Price et al., 1994). According to the SST observations from the satellite, SST usually decreases 2-5 ℃ due to typhoon passing (Cione and Uhlhorn, 2003; D'Asaro et al., 2007; Wu et al., 2008; Jiang et al., 2009). Dare and Mcbride (2011) studied the response of SST to the global typhoons during 1981~2008 and indicated that the maximum decreasing of SST usually occurred in 1-day after typhoon passing.

In this section, we selected the typhoon Tembin as an example to validate the MO and SCSOFS model skills for the SST simulations. As shown in Fig. 11, the typhoon Tembin passed through and made a perfect turn  in the NSCS from 25 to 28 August 2012. From the three results, we can find the obvious decreasing of SST 1-day after typhoon passing, which is about 2-4 ℃ and in good correspondence with previous studies mentioned in above. SCSOFS is in much better agreement with OISST than MO, especially on 26 and 27 August 2012, not only for the range of SST decreasing, but also for the domain of SST decreasing.

**3.3.3 Mesoscale eddy**

Mesoscale eddies cannot be identified and extracted from geophysical turbulent flow as observed by satellite altimetry without suitable definition and a competitive identification algorithm. A number  of different techniques for automatic identification of eddies have been proposed based either on physical or geometric criteria of the flow field. In this study, a free-threshold eddy identification algorithm with the SLA data is employed. This algorithm is based on the vector geometry method and Okubo-Weiss method (Okubo, 1970; Weiss, 1991) with six constraints applied to the SLA to detect an eddy: (1) a vorticity-dominated region at the eddy center (W < 0**, here W is the Okubo-Weiss parameter, for its definition referred as Xiu et al.(2010)**) must exist; (2) the SLA magnitude has a local extreme value (minimum or maximum); (3) closed contours of SLA around the eddy center must exist; (4) the eddy radius must be larger than 45km.(5)the eddy amplitude must be larger than 4cm. In this study, the amplitude is defined as the absolute value of the SLA difference between the eddy center and the SLA along the eddy edge. The Eddy-tracking method used is the one developed by Chaigneau et al. (2008), and we only keep eddies with life span not less than 28 days. Eddies were analyzed and compared based on MO, AVISO and SCSOFS in 2012. The numbers of eddies for three types of data were in Table 1, cyclones and anti-cyclones were counted separately and seasonally.

**Comment [ZA4]:** What is this?

The spatial distribution of eddy originbirthplace is shown in Fig 12. MO has more eddies formed, especially more anti-cyclones formed than those inof AVISO, most of the excessive eddy cores were found near the middle of SCS. SCSOFS has more anti-cyclones as well and less cyclones than AVISO. Both MO and SCSOFS show excessive eddies formed in the middle of the basin and less eddies in the western of the east of Vietnam. The SLA of SCSOFS and MO is calculated simply by subtracting mean SSH (214 years mean for SCSOFS and only one-year mean for MO) instead of an uniformed Mean Sea Surface, which might cause the excessive anti-cyclones in both models. Observations of AVISO, and model results from MO and SCSOFSAll three types of data show agree that less eddies formed in the middle part of NSCS.

As for the seasonal distributions (figures not shown), all three data have most eddies in spring and lest in fall. Both AVISO and SCSOFS have lest eddies in fall, and more cyclones than anti-cyclones in spring and fall, and all three have less cyclones than anti-cyclones in summer. SCSOFS differs with AVISO mainly in winter while they agree reasonably in the other three seasons. MO has surplus eddies counted in every season especially for anti-cyclones, which might be causes of the errors introduced by the simplified calculation of SLA.

**4 Conclusions**

Two operational ocean analysis and forecasting systems, MO and SCSOFS, have been built based on the state-of-the-art hydrodynamic ocean model in France and China, respectively. This paper demonstrated the results of comparison and validation for the performance of both systems on the ocean circulation, the structures of the TS, and mesoscale activities in the SCS, based on the observed satellite and *in-situ* data in 2012, are shown in this paper. The comprehensive performances for the both systems are summarized as follow.

Both systems have capabilities to simulatecaught the main basin-scale circulations in the SCS and model results are in goodbeen well agreement with the result of AVISO data. And MO has better performance than SCSOFS in simulatingthe results of MO are better agreement with those of AVISO than those of SCSOFS for several branches and eddies in January. SCSOFS did not generateThere are no many mesoscale or small scale circulations shown in SCSOFS, which may be caused byof a little strong horizontal mixing set in the model. The westward intensification in the eastern coast of the Vietnam is

> **Comment [ZA5]:** What does this refer?

most strong in autumn among the four seasons. For the type of the Kuroshio intruding the SCS, the AVISO observations, and model  results from both MO and SCSOFS show the looping path in winter, the leaping path in summer and leaking path in autumn. However, the leaking path, looping path and leaping path are shown for AVISO, MO and SCSOFS in spring.

Both systems demonstrated  the same variation trends in u and /v- components time series with the mooring observation. The RMSE between MO, SCSOFS and mooring observation are 0.075 m/s, 0.094 m/s for u-component, 0.062 m/s, 0.084 m/s for v-component, respectively. The results of MO are in better agreement with the observation than those of SCSOFS, especially during the period of the Typhoon Kai-tak.

The maximum, minimum, and mean for the basin-averaged 12 monthly RMSEs between MO and MGDSST are 0.78 °C, 0.37 °C, 0.51 °C, between SCSOFS and MGDSST are 1.15 °C, 0.56 °C, 0.86 °C in the SCS, respectively. For the horizontal and vertical distributions of TS, both systems have achieved  the same structures with the *in-situ data*, but the results of SCSOFS are in better agreement with the *in-situ* observations than those of MO. The correlation coefficients  are 0.987 and 0.982 for temperature,  0.717 and 0.897 for salinity, between model results from MO and SCSOFS and *in-situ data*, over the 95% significance level, respectively. It indicates that the good relativity between MO, SCSOFS and *in-situ*, the relativity of temperature is better agreement with *in-situ* than those of salinity for both MO and SCSOFS, and SCSOFS is better agreement with *in-situ* than MO for salinity.

The similar SST frontal patterns with their evident seasonal variations are shown in both systems. Most fronts achieve maximum strength in winter, become weak obviously due to the weakening of northeaster monsoon EAM in spring and summer, fade and disappear in autumn, which is consistent with the result of Wang et al. (2001).

During the typhoon Tembin in the NSCS, the obvious decreasing of SST about 2-4 °C occurs 1 day after typhoon passing shown in the results of MO, SCSOFS and OISST, which is consistent with previous studies. SCSOFS is in much better agreement with OISST than MO for both  range and domain of SST decreasing.

MO has more eddies formed near the middle of SCS than AVISO, especially for anti-cyclones. SCSOFS has more anti-cyclones, but less cyclones than AVISO. AVISO data and model results from NO and SCSOFS all  show most eddies in spring and lest in fall, and less cyclones than anti-cyclones in summer. Both AVISO and SCSOFS have more cyclones than anti-cyclones in spring and fall.

In order to improve  performances of MO and SCSOFS in the SCS in future based on the results of  the comparison and validation for the two systems,

Some recommendations are proposed as below:

For MO, we would like to suggest (1) to modify the model bathymetry in the coast area for the depth less than 200m to improve the model performance in shallow water area, such as SST front; (2) to change the initial conditions of TS to improve the TS vertical structures, especially for the salinity in deep water area; For SCSOFS, we would like to suggest (1) to weaken horizontal mixing to get more reasonable mesoscale or small scale circulations; (2) to optimize the data assimilation scheme further to better assimilate the *in-situ* and satellite data; (3) to replace the surface forcing data with the higher horizontal or temporal resolution; (4) to replace the boundary conditions from monthly to weekly or daily like

MO. For both systems, we also would like to suggest to try to get and assimilate more observed data during the typhoon period to catch the typhoon process more exactly.

**Author contribution**

X. Zhu, H. Wang and G. Liu compared and validated the model results on velocities and TS. C.

Régnier and M. Drévillon build the MO, D. Wang build the SCSOFS. X. Kuang analyzed the model results on mesoscale eddy. S. Ren analyzed the model results on SST front. Z. Jing provided the *in-situ*

data. X. Zhu prepared the manuscript with contributions from all co-authors.

**Acknowledgements**

We would like to thank the anonymous reviewers and the Editor,    , for their valuable contributions that allow us to improve the manuscript substantially. This study is supported by the National Natural

Science Foundation of China under contract No. 41222038, 41376016, 41206023 and the

Strategic Priority Research Program of the Chinese Academy of Sciences Grant No. XDA1102010403.

[revised manuscript text omitted]

---

## Author Comment (AC2) · 8 May 2016

**1. General comments**

This paper assessed and validated the performance of two operational ocean forecasting systems, the global Mercator Ocean operational forecast system developed in France and the regional South China Sea operational forecasting system developed in the National Marine Environmental Forecasting Center in China by comparing the model results from the two systems with a comprehensive set of observations from satellite and in-situ measurements. The recommendations were proposed for future improvement of both systems based on comparison results. The methodology and matrix of evaluation and validation are reasonable, the observations are well quality controlled. My detail edits were added in the manuscript with review "track changes"

[Figure]

and sent the revised Microsoft word document back to the authors. I would recommend accept this paper for publication after further editing of grammars.

We are really appreciate the reviewer for the constructive comments and detail edits of grammars. We answered all the comments and revised the manuscript to significantly improve the manuscript in the revision as follow:

(1) Line 260: What are three results referring? Results from MO and SCSOFS, what is the third? (geostrophic flow?)

The three results refer the observations of AVISO, the model results from MO and SCSOFS. We have changed the text at line 259 of the revised manuscript.

(2) Line 332: It is not clear which layer, "Fig.7 for the layer above 300 m and Fig.8 for the layer of 300-1200m"

Figure 7 shows the vertical distribution of TS at the layer of depth shallower than 300 m, Fig. 8 shows the vertical distribution of TS at the layer of depth from 300 m to 1200 m. We have changed the text at line 331 of the revised manuscript.

(3) Line 399: What "A number multitude" is this?

Sorry, there is a typo over there. We have removed the word "multitude" at line 400 of the revised manuscript.

(4) Line 437: What does "several branches" refer?

It means the main branches of the SCS ocean circulations. We have changed the text at line 438 of the revised manuscript.

(5) Line 455: It is not clear for "It indicates that the good relativity between MO, SCSOFS and in-situ, the relativity of temperature is better agreement with in-situ than those of salinity for both MO and SCSOFS, and SCSOFS is better agreement with in-situ than MO for salinity.

We have pointed out that "the correlation coefficients are 0.987 and 0.982 for temperature, 0.717 and 0.897 for salinity, between model results from MO and SCSOFS and in-situ data, over the 95% significance level, respectively". The correlation coefficients are over 0.7, they can indicate the good relativity between the model results and in-situ observations. The correlation coefficients of temperature, 0.987 and 0.982 between model results and in-situ observations, are larger than the correlation coefficients of salinity, 0.717 and 0.897. It indicates that the "the relativity of temperature is better agreement with in-situ than those of salinity for both MO and SCSOFS". The correlation coefficient (0.897) of salinity between SCSOFS and the in-situ observations is larger than the one (0.717) between MO and the in-situ observations. It indicates that "the result from SCSOFS is better agreement with in-situ data than MO for salinity".

Please also note the supplement to this comment:
http://www.nat-hazards-earth-syst-sci-discuss.net/nhess-2016-60/nhess-2016-60-AC2-supplement.pdf

––––––––––––––––––––––––––––––––

[Figure]

**Supplement:**

**Comparison and validation of global and regional ocean forecasting systems for the South China Sea**

Xueming Zhu[1], Hui Wang[1], Guimei Liu[1*], Charly Régnier[2], Xiaodi Kuang[1], Dakui Wang[1], Shihe Ren[1], Zhiyou Jing[3], Marie Drévillon[2]

[1]Key Laboratory of Research on Marine Hazards Forecasting, National Marine Environmental Forecasting Center, Beijing, 100081, China

[2]Mercator Océan, Ramonville Saint Agne, France

[3]State Key Laboratory of Tropical Oceanography, South China Sea Institute of Oceanology, Chinese Academy of Sciences, Guangzhou, 510301, China

*Correspondence to*: Guimei Liu (liugm@nmefc.gov.cn)

**Abstract.** In this paper, the performance of two operational ocean forecasting systems, the global Mercator Océan (MO) Operational System, developed and maintained by Mercator Océan in France and the regional South China Sea Operational Forecasting System (SCSOFS) by the National Marine Environmental Forecasting Center (NMEFC) in China, have been examined. Both systems can provide science-based nowcast/forecast products of temperature, salinity, water level and ocean circulations. comparison and validation of the ocean circulations, the structures of the temperature and salinity, and some mesoscale activities, such as ocean fronts, Typhoon, and mesoscale eddy, are conducted  based on the observed satellite and *in-situ* data obtained in 2012 in the South China Sea,. The results showed that MO performs better in simulating the ocean circulations and SST , and SCSOFS.  performs better in simulating the structures of temperature and salinity . For the mesoscale activities, SCSOFS performance is better than MO in simulating SST fronts and SST decreasing during the typhoon Tembin  compared with the previous studies and satellite data ; but model results from both of SCSOFS and MO show some differences from satellite observations. on In conclusion, some recommendations have been proposed for both forecast systems to improve their forecasting performance in the near further based on our comparison and validation.

**Keywords.** SCSOFS, Mercator Océan, South China Sea, Operational Forecasting System

**1 Introduction**

The South China Sea (SCS, Fig.1) is the largest and deepest semi-enclosed marginal sea of the Northwest Pacific Ocean (NWP), with the area about 3.5 million km$^2$, the mean and maximum depth about 1200 m and 5300 m, respectively. The northern SCS (NSCS) is a wide continental shelf with depth less than 200 m  and the southern SCS (SSCS) is a large basin, comprising.  numerous islands, reefs, beaches, shoals . SCS is connected with the adjacent seas through a number of channels, to the East China Sea in north, to the NWP in east, to the Sulu Sea in southeast, and to the Java Sea in south, by the Taiwan

Strait (TWS), the Luzon Strait (LUS), the Mindoro Strait and the Balabac Strait, the Karimata Strait, respectively. SCS has Iits unique geographical features, rich marine mineral and petroleum resources so that it is very important forplay a significant role to many countries adjacent to it.

The SCS is located in the East Asian Monsoon (EAM) winds regime, the northeasterly winds usually prevail with an average wind speed of 9 m/s over the whole domain in winter, while the southwesterly winds prevail with an average magnitude of 6 m/s dominating over the most parts of the SCS in summer (Hellerman and Rosenstein, 1983). The EAM is considered to be the majorin factors for driving the upper layer basin-scale circulation pattern in the entire SCS, showing an obvious seasonal variation with a cyclonic gyre in winter and an anti-cyclonic gyre in summer (Wyrtki, 1961; Mao et al., 1999; Wu et al., 1999; Qu, 2000; Chu and Li, 2000). However, some other literatures insist that a persistent cyclonic gyre is present in the NSCS, while a semibiannually changeing from a cyclonic gyre in winter to an anti-cyclonic gyre regime in summer can be observed 
[revised manuscript text omitted]
's infrared sensors (AVHRR/NOAA) and microwave sensor (AMSR-E/AQUA), and *in-situ* SST (buoy and ship) data Global Daily SST (MGDSST), with a 0.25°×0.25° horizontal resolution, which is analyszed and published at the Office of Marine Prediction of theby Japan Meteorological Agency (JMA). The data can be obtained from http://near-goos1.jodc.go.jp/.

The other one is derived from the NOAA 0.25°×0.251/4° daily Optimum Interpolation Sea Surface

Temperature (OISST), which is an analysis constructed by combining observations from different platforms, such as satellites, ships, buoys, on a regular grid via optimum interpolation. Right now,

National Centers for Environmental Information (NCEI) provides two kinds of OISST: one uses infrared satellite data from the Advanced Very High Resolution Radiometer (AVHRR) named as AVHRR-only, and the other one uses AVHRR data along with microwave data from the Advanced Microwave

Scanning Radiometer (AMSR) on the Earth Observing System Aqua or AMSR-E satellite named as

AVHRR+AMSR. Since the production of the AVHRR+AMSR data ended in 2011, tThe first one,

AVHRR-only, is used in this study, which spans 1981 to the present and can be downloaded from the website http://www.ncdc.noaa.gov/oisst/data-access.

**2.2 *In-situ* data**

The *in-situ* data employed in this paper for the comparison and validation of both systems are provided by the South China Sea Institute of Oceanology, Chinese Academy of Sciences. There were one mooring to measure the sea water velocity and 5 cruises conductedimplemented to measure the temperature and salinity (TS) in the SCS during 2012.

The mooring station is located at Maoming (Fig. 1), where bottom-mounted upward-looking 75 kHz

Acoustic Doppler Current Profilers (ADCPs) wereare deployed to monitor the current profile (U

component and V component) from the depth of 2 m to 48 m with a 2-m interval in vertical. The period of the monitoring is from 11 July to 8 October, in 2012, with a temporal interval 10 min. Firstly, the outlierabnormal data are eliminated from the original measured data; in the secondly, a low-pass filter with 25-hour is applied to removefilter out the tidal currents; and daily mean currents are calculated using a 25-hour averaginge is calculated to get the daily average data, which were used in order to compare and validate with the simulated results of MO and SCSOFS.

The TS data from the five5 cruises wereare measured by SeaBird 19 plus conductivity-temperature-depth (CTD) with 1-m resolution in vertical. Among the five5 cruises, one is the Qiongdong cruise in the NSCS, which was conducted for 9 days from 12 to 20 July at 90 stations along 6 sections (See Fig.1); another one is the Nansha cruise around the Nansha Islands, which was conducted for 5 days from 24 to 28 August at 17 stations along 10°N section from 109.5°E to 117.5°E.

The TS data from those two cruises will be used to compare and validate the TS distribution from MO

and SCSOFS in vertical and horizontal.  The TS data collected   from all the 5 cruises will be used to perform a correlation analysis of each of the simulated predictions of

MO and SCSOFS models with the obervations.

**2.3 The configurations of SCSOFS**

The SCSOFS uses the Regional Ocean Modeling System (ROMS), which is a three-dimensional, non-linear primitive equations, free surface, hydrostatic, split-explicit, topography-following-coordinate in vertical and orthogonal curvilinear in horizontal on a staggered

Arakawa C-grid (Arakawa and Lamb, 1977) oceanic model (Shchepetkin and McWilliams, 2005).

To avoid the influences of boundary to the circulations in the SCS, the model's boundaries was extended to southward and eastward, then the model covered a larger domain (4.5°S to 28.3°N, 99°E to 145°E, Fig.

1) than the SCS. The horizontal resolution variates from 1/12° in the south and east boundary to 1/30° in the SCS. There were 36 s-coordinate levels in the vertical with the thinnest layer being 0.16 m on the surface. The bathymetry was extracted from the ETOPO1 data sets published by U.S. National

Geophysical Data Center (NGDC), which is a global relief model of Earth's surface that integrates ocean bathymetry and land topography, with 1 arc-minute resolution (Amante and Eakins, 2009). The ETOPO1

data set has combined the satellite altimeter observations, shipping load sonar measurement, multi resolutions digital terrain database and the global digital terrain model and many other data sources, and it has been used in the global and regional oceanic models widely. And the original ETOPO1 bathymetry was revised in the area of near the coast of China mainland according to the *in-situ* data collected in NMEFC, then smoothed according to Shapiro (1975). The maximum depth was set to be 6000 m and the minimum depth to be 10 m in the model (Wang, 1996).

The initial temperature and salinity conditions were derived from the climatology monthly mean (SODA) in January. However, the initial velocities and elevation were set to zero, which means to integrate the model from a static status. The model's western lateral boundary was treated as a wall. The other three (northern, southern, eastern) lateral boundaries were opened, whose temperature, salinity, velocity, and elevation were prescribed by spatial interpolation of the monthly mean SODA dataset. The 2D and 3D velocities, through the open boundaries, are modulated to guarantee the conservation of volume flux in the whole model domain. In addition, the nudging technology was used for 3D velocity, temperature, and salinity to the three open lateral boundaries with a 30-day time scale for outflow and 3-day for inflow.

The model is forced using 6-hourly wind stress, net fresh water fluxes, net heat fluxes, surface solar shortwave radiation at surface from NCEP_Reanalysis 2 data provided by the NOAA/OAR/ESRL PSD,

Boulder, Colorado, USA, accessible from their web site at http://www.esrl.noaa.gov/psd/ (Kanamitsu et al., 2002). In order to get more reasonable simulated SST, the kinematic surface net heat flux sensitivity to SST (dQ/dSST) is used to introduce thermal feedback to correct net surface heat flux (Barnier et al.,

1995) with a constant number -30 $W/m^2/°C$ in the whole domain. The MGDSST data is used to correct net surface heat flux. In addition, the monthly mean climatology discharges of the Mekong River and the

Pearl River are prescribed to the model.

The system was run with 6 seconds time step for the external mode, and 180 seconds for the internal mode under the initial conditions, boundary conditions and surface forcing mentioned in above. The system was conducted a hindcast run from 2000 to 2011 after a 15 years climatology run for spin-up (Wang et al., 2012). The model results wereare archived to the snapshot with a 5-day interval, which werewill be used as the ensemble members for the EnOI (Ensemble Optimal Interpolation) method assimilation. After the hindcast run, the system was conducted an assimilation run in 2012 with EnOI

method, the along track SLA data from AVISO had been assimilated as the observations with a 7-day time window. The details on the EnOI applied in the SCSOFS can be referred as Ji et al. (2015). The assimilated results wereare archived to daily mean with a 1-day interval in 2012, which werewill be used to compare and validate in this paper. Then the system is implemented into operationsng in NMEFC

since January 1st, 2013. It runs on daily bases for 6 days simulations (1-day nowcast and 5-day forecast)

to, and provides 120-hour forecasting products, which including ofthe three dimensional 3D ocean temperature, salinity and currents with 24 hours interval.

**2.4 The configurations of MO**

The high resolution global analysis and forecasting system PSY4V1R3 was operational as the V2 of the

MyOcean project from February 2011 up to April 2013, when it was replaced by the PSY4V2R2 system.

During this period, PSY4V1R3 has been producing weekly 14-day hindcasts and daily 7-day forecasts.

**The PSY4V1R3 configuration described as followed is indicated for as MO model through this paper.**

The model configuration of PSY4V1R3 is based on a tripolar ORCA grid type (Madec and Imbard, 1996)

in the NEMO 1.09 version with a 1/12° horizontal resolution which means 9 km at the Equator, 7 km at mid latitudes and 2km toward the Ross and Weddel Sea. The grid cells follow an Arakawa C-grid type (Arakawa and Lamb, 1977). The 50-level vertical discretization retained in this system has 1m resolution at the surface, decreasing to 450 m at the bottom and 22 levels within the upper 100 m. "Partial cell"

parametrization was chosen for a better representation of the topographic floor (Barnier et al., 2006). The high frequency gravity waves are filtered out by the free surface formulation of Roullet and Madec (2000).

For the diffusion, a horizontal bilaplacian was added along the equator ($20 m^2s^{-1}$) and two laplacians in the Canadian straits (up to $100 m^2s^{-1}$). Laplacian lateral isopycnal diffusion was added on tracers (125

$m^2s^{-1}$) and a horizontal biharmonic viscosity was added for the momentum ($-1\times1010 m^4s^{-1}$ at the

Equator and decreasing poleward as the cube of the grid size). In addition, the vertical mixing is parameterized according to a turbulent closure model (TKE order 1.5) adapted by Blanke and Delecluse (1993), the lateral friction condition is a partial-slip condition with a regionalization for the

Mediterranean Sea, Indonesian region, Canadian straits and Cape Horn. The atmospheric fields are taken from the ECMWF (European Centre for Medium Range Weather Forecasts) Integrated Forecast System at a daily average frequency. Momentum and heat turbulent surface fluxes are computed from CLIO bulk formulae (Goosse et al., 2001). A viscous-plastic rheology formulation is used for the LIM2_VP

ice model (Fichefet and Maqueda, 1997, LIM2_VP in Hunke and Dukowicz 1997). A multivariate data assimilation (Kalman Filter kernel with SEEK formulation , Pham et al., 1998) of *in-situ* T and S (from

Coriolis/Ifremer), along--track MSLA (from AVISO, with MDT from Rio and Hernandez, 2004) and intermediate resolution SST (0.25°×0.25 SST product RTG from NOAA) is performed with the

SAM2 software (Lellouche et al., 2013). An Incremental Analysis Update (IAU) centered on the 4th day of the 7-day assimilation window ensures a smooth correction of T, S, U, V and SSH (Sea Surface

Height). The assimilation cycle consists of a first 7-day simulation called guess or forecast, at the end of which the analysis takes place. The IAU correction is then computed and the model is re-run on the same week, progressively adding the correction. The increment is distributed in time with a Gaussian shape which is centered on the 4th day. More details on the SAM2 software (applied on other model configurations) can be found in Lellouche et al. (2013) except that no large scale bias correction is applied in PSY4V1R3. Concerning the initial conditions, the PSY4V1R3 was started in April 2009 from a 3D climatology of temperature and salinity (WOA2005, Antonov et al., 2006; Locarnini et al., 2006).

**3 Comparisons, validations and discussion**

**3.1 Velocities**

**3.1.1 Absolute Geostrophic Velocity**

Figure 2 shows the distributions of the monthly AGV composited with SSH from

AVISO, MO, and SCSOFS in January, April, July, and October of 2012, respectively. Here we use the

January, April, July, and October represent winter, spring, summer, and autumn, respectively. It is valuable to note that the AGV of MO and SCSOFS are not the velocities output from the numerical model directive. However, in order to better comparison, they are recalculated according to SSH from the model output on every day and assuming geostrophic balance following Eq. (1):

$$u = -\frac{g}{f}\frac{\partial SSH}{\partial y} \qquad v = \frac{g}{f}\frac{\partial SSH}{\partial x} \qquad\qquad (1)$$

where $g$ is gravitation acceleration, $f$ is the Coriolis parameter, $x$, $y$ are the east, north axis; $u$, $v$ are the eastward, northward velocity components in horizontal, respectively.

Comparisons of the observations of AVISO with the results from MO and SCSOFS shows that both MO and SCSOFS can catch the main basin-scale oceanic circulation pattern in the SCS, and show that a cyclonic gyre in winter and an anti-cyclonic gyre in summer, which being well accordance with the pattern of AVISO. It is worth to mention that the result of MO is in good agreement with the AVSIO in January, such as the southward western boundary currents along the eastern coast of Vietnam, the LCC, the anti-cyclonic eddy in the western of the LUS around (118°E, 21°N), the cyclonic eddy in the eastern of the Vietnam around (113°E, 15°N). However, the result of SCSOFS is much smoother without obvious mesoscale or small scale circulations, or they are very weaker (0.2-0.4 m s$^{-1}$) than those (0.6-0.8 m s$^{-1}$) of AVISO or

MO. The circulation is chaos in spring in the SCS, though the circulation pattern of MO is in better agreement with the one of AVISO than the one from SCSOFS.  The anti-cyclonic eddy around (111°E, 10°N) and the western boundary jet in the southeast of the Vietnam in summer, with the maximum speed being about 1.0 m s$^{-1}$, 0.9 m s$^{-1}$, and 0.7 m s$^{-1}$ are shown by AVISO,

MO, SCSOFS, respectively. The westward intensification along the eastern coast of the Vietnam is morest obvious in autumn than other three seasons, and the maximum speed is larger than 1.0 m s$^{-1}$

for MO and SCSOFS, but is about 0.7 m s$^{-1}$ for AVISO.

As mentioned in Sect. 1, the Kuroshio intruding the SCS through the LUS has been distinguished by three types as the looping path, the leaking path and the leaping path, according to Nan et al. (2011a). All three results show the looping path in winter, the leaping path in summer and leaking path in autumn, which is well consistent with the model results showed by Wu and Chiang (2007). However,

AVISO, MO, and SCSOFS show the leaking path, looping path, and leaping path in spring, respectively.

**3.1.2 Time series from mooring station**

Figure 3 shows the comparison of the daily mean time series of  u, v components from the mooring,

MO, and SCSOFS at 40m-depth layer at the Maoming station (See Fig. 1) from July 11 to October 8,

2012. Both MO and SCSOFS can capture the similae variation trends of the time series with the mooring observation. Especially, MO results match the observed current variations well for both u- and v- component, during the period of the Typhoon Kai-tak on 17 August 2012. Although

SCSOFS shows the larger velocity during the Typhoon Kai-tak, the maximum velocity range is less than the observation and anticipating  the observation about 1 day. The root mean square errors (RMSE) between observations and models of MO and SCSOFS are 0.075 m s$^{-1}$,

0.094 m s$^{-1}$ for u-component, 0.062 m s$^{-1}$, 0.084m s$^{-1}$ for v-component, respectively. Overall, MO results are in better agreement with the observations than  SCSOFS.  SCSOFS results have a temporal bias (phase shift) comparing with the observation, which is leading the observations about

1 day.

**3.2 Temperature and Salinity**

**3.2.1 SST**

SST is a very important prognostic variable in a hydrostatic ocean general circulation numerical model, which plays a key role to the ocean circulations and the air-sea interaction. So SST error is a crucial criteria of the numerical model skill, especially for an operational ocean circulation model. In fact, the

SST simulation error is affected by several factors, for example the limitation of physical model, the surface atmospher forcing conditions, the bias of initial field and the uncertainty from the open boundary, as pointed out by Ji et al. (2015). Although the SST data have been assimilated into both MO and SCSOFS, the assimilated SST still has some errors for both systems.

Figure 4 shows the distributions of the monthly mean SST errors between two systems and MGDSST in the SCS in 2012. The errors show an obvious regional distribution, the bigger errors mainly appear in the coastal regions for the depth shallower than 200 m, such as in the TWS, the eastern of the Guangdong province in January, the gulf of Tonkin in July. Tthe strong seasonal variations for the basin-averaged SST error can also be found, which is larger in winter and smaller in summer for both systems. Comparing with MGDSST, the maximum, minimum, and mean for the basin-averaged 12 monthly RMSEs are 0.78 ℃, 0.37 ℃, 0.51 ℃ for the MO, 1.15 ℃, 0.56 ℃, 0.86 ℃ for the SCSOFS, respectively, in the SCS. Based on the Fig. 4, MO performed better than SCSOFS in simulating SST  by comparing with MGDSST.

**3.2.2 Horizontal and vertical distribution of TS**

TS horizontal distributions at 10-m depth layer  in the eastern of Hainan island from the *in-situ* observations of Qiongdong cruise, model results from MO, and SCSOFS are shown in Fig. 5. Two clear cold and salty water cores located at the eastern of Hainan island, which are located at about (110.75°E, 19.2°N) and (111.3°E, 19.7°N), are shown in both *in-situ* observations and SCSOFS (Fig. 5) with the cores from SCSOFS being more saline than the *in-situ observations*. It can be easily deduced that the two cores are produced by upwelling process from the TS vertical distributions of the section K, F, H, and G (Jing et al., 2015).

Figure 6 shows the vertical TS distributions from the *in-situ* observations of Qiongdong cruise, model results from MO and SCSOFS, along section E. Both systems have gotten the same vertical structures of TS with the *in-situ observations*. All of them demonstrated the obvious upwelling systems, with cold and salty waters flowing from offshore to nearshore along the bottom. All three results show the upper mixing layer depth is about 15 m.the sea water is well mixed above 15 m depth and the isotherms and isohalines are almost vertical, where strong vertical stratification is shown in summer. The diluted water is flushing from the nearshore to offshore, with the 33-isohaline cross with the sea surface  at the position of about 50 km from the coast for both *in-situ*

observations and SCSOFS, but at the position of about 420 km for MO. In above, it is indicated that the results of SCSOFS is better agreement with the *in-situ* observations than those of MO.

The vertical distributions of TS from the *in-situ* observations of Nansha cruise, and model results from MO, and SCSOFS along the 10°N section are shown in Fig.7 for the layer of depth shallower than above 300 m and Fig.8 for the layer of depth from 300 m to 1200 m. Both systems have got almost the same vertical structures with the *in-situ*, especially for the upper mixing layer depth about 70 m are shown in the three results. Water The temperature almost linearly decreases from 28 ℃ to 3 ℃ with the depth going deep increasing from the bottom of the upper mixing layer to the 1200 m depth. However, the salinity increases from 33.5 psu to 34.5 psu with the depth going deep increasing from the bottom of the upper mixing layer to about 200m depth, and keeps almost constant at 34.5 psu from 200 m to 300 m depth. Then a fresher water layer exists in the middle layer from about 400 m to 700 m with the salinity about 34.4 psu. Below the middle layer, the salinity again increases from 34.4 to 34.58 with the depth increasing from 700 m to 1200 m. It indicates that the results of MO and SCSOFS are in good well agreement with *in-situ observations*, except that the salinity of the fresh water in the middle layer from MO is less than 34.4 which and is fresher than those of *in-situ* and SCSOFS, but the thickness of the fresh layer is thicker than those of *in-situ* and SCSOFS.

**3.2.3 Correlation analysis ship between model and *in-situ***

In order to better compare and validate the performances of the two systems, we collected all the measured TS data from five cruises in the SCS in 2012 to conduct a comprehensive correlation analysis. Figure 9 shows the comparison of relativity of TS between the model results from MO and, SCSOFS and the *in-situ* observation by scatter points, respectively. Each Any point in the Fig.9 is corresponded sd with two values of temperature or salinity, one is from the *in-situ* observation along X axis, and the other one is from the model results from MO or SCSOFS along Y axis. The correlation coefficients of temperature are 0.987, 0.982, and of salinity are 0.717, 0.897, between MO, SCSOFS and *in-situ*, over the 95% significance level, respectively, which is showing the good relativity between MO, SCSOFS and *in-situ*. It also indicates that the relativity of temperature is in better agreement with *in-situ* than those of salinity for both MO and SCSOFS, and SCSOFS is in better agreement with *in-situ* than MO for salinity.

**3.3 Mesoscale activities**

**3.3.1 SST front**

Oceanic front is a good indicator for connection between water masses with different hydrological features, which is an important marine mesoscale phenomenon. There are numerous SST fronts in the SCS, most of them located on the continental shelf with the depth below 200 m or aligned with the shelf break, especially in the NSCS. A few obvious SST fronts have been identified from the long-term NOAA/NASA Pathfinder SST data, namely: Fujian-Guangdong Coastal Front, Pear River Estuary Coastal Front, Taiwan Bank Front, Kuroshio Intrusion Front, Hainan Island East Coastal Front, Tonkin Gulf Coastal Front (Wang et al., 2001). All of them exhibit very strong seasonal variability, which is mainly due to the EAM (Belkin and Cornillon, 2003).

Figure 10 shows the distributions of SST fronts from the model results from MO and SCSOFS in four seasons. The similar frontal patterns with their evident seasonal variations are shown in both systems, except for  small differences. In winter, most fronts reach maximum strength (>0.2 ℃/km). The Fujian-Guangdong Coastal Front and Taiwan Bank front are major fronts in the SCS which is in agree with previous satellite result from Wang et al. (2001). These two fronts merge and extend to Pearl River Estuary and the Hainan Island. The Hainan Island East Coastal Front is stronger in MO than in SCSOFS, whereas the Tonkin Gulf Coastal Front is stronger in SCSOFS than in MO.  the Kuroshio Intrusion front is obvious in SCSOFS, however, it is hardly seen in MO. In spring, most fronts become weak obviously due to the weakening of northeast monsoon fo both operational systems, except that the Hainan West Coastal front emerges in SCSOFS. In summer, weakening almost occurs in all the fronts mentioned above for SCSOFS, which is in good agreement with the result of Wang et al. (2001). However, disappearing occurs in all the fronts for MO. In fall, most fronts fade and disappear, except that the Taiwan Bank front has very weak strength compared to other seasons for both systems. Both systems have not shown the Kuroshio Intrusion Front identified by Wang et al. (2001) in summer and fall.

**3.3.2 The Typhoon Tembin**

There are a lot of typhoons in the SCS during the typhoon season in every year, so that the typhoon activities are very frequent in the SCS, especially in 2012. One important study on the air-sea interaction is the responding of the physical ocean dynamics to typhoon in the oceanic upper layer. One important responding is the decreasing of SST due to the strong vertical mixing caused by typhoon (Price et al., 1994). According to the SST observations from the satellite, SST usually decreases 2-5 ℃ due to typhoon passing (Cione and Uhlhorn, 2003; D'Asaro et al., 2007; Wu et al., 2008; Jiang et al., 2009). Dare and Mcbride (2011) studied the response of SST to the global typhoons during 1981~2008 and indicated that the maximum decreasing of SST usually occurred in 1-day after typhoon passing.

In this section, we selected the typhoon Tembin as an example to validate the MO and SCSOFS model skills for the SST simulations. As shown in Fig. 11, the typhoon Tembin passed through and made a perfect turn  in the NSCS from 25 to 28 August 2012. From the three results, we can find the obvious decreasing of SST 1-day after typhoon passing, which is about 2-4 ℃ and in good correspondence with previous studies mentioned in above. SCSOFS is in much better agreement with OISST than MO, especially on 26 and 27 August 2012, not only for the range of SST decreasing, but also for the domain of SST decreasing.

**3.3.3 Mesoscale eddy**

Mesoscale eddies cannot be identified and extracted from geophysical turbulent flow as observed by satellite altimetry without suitable definition and a competitive identification algorithm. A number[ZA1] of different techniques for automatic identification of eddies have been proposed based either on physical or geometric criteria of the flow field. In this study, a free-threshold eddy identification algorithm with the SLA data is employed. This algorithm is based on the vector geometry method and Okubo-Weiss method (Okubo, 1970; Weiss, 1991) with six constraints applied to the SLA to detect an eddy: (1) a vorticity-dominated region at the eddy center ($W < 0$, here **W is the Okubo-Weiss parameter, for its definition referred as Xiu et al.(2010)**) must exist; (2) the SLA magnitude has a local extreme value (minimum or maximum); (3) closed contours of SLA around the eddy center must exist; (4) the eddy radius must be larger than 45km.(5)the eddy amplitude must be larger than 4cm. In this study, the amplitude is defined as the absolute value of the SLA difference between the eddy center and the SLA along the eddy edge. The Eddy-tracking method used is the one developed by Chaigneau et al. (2008), and we only keep eddies with life span not less than 28 days. Eddies were analyzed and compared based on MO, AVISO and SCSOFS in 2012. The numbers of eddies for three types of data were in Table 1, cyclones and anti-cyclones were counted separately and seasonally.

The spatial distribution of eddy origin is shown in Fig 12. MO has more eddies formed, especially more anti-cyclones formed than those in AVISO, most of the excessive eddy cores were found near the middle of SCS. SCSOFS has more anti-cyclones as well and less cyclones than AVISO.

Both MO and SCSOFS show excessive eddies formed in the middle of the basin and less eddies in the western of the east of Vietnam. The SLA of SCSOFS and MO is calculated simply by subtracting mean

SSH (14 years mean for SCSOFS and only one-year mean for MO) instead of an uniformed Mean Sea

Surface, which might cause the excessive anti-cyclones in both models. Observations of AVISO, and model results from MO and SCSOFS show  that less eddies formed in the middle part of NSCS.

As for the seasonal distributions (figures not shown), all three data have most eddies in spring

. Both AVISO and SCSOFS have lest eddies in fall, and more cyclones than anti-cyclones in spring and fall, and all three have less cyclones than anti-cyclones in summer. SCSOFS differs with AVISO

mainly in winter while they agree reasonably in the other three seasons. MO has surplus eddies counted in every season especially for anti-cyclones, which might be causes of the errors introduced by the simplified calculation of SLA.

**4 Conclusions**

Two operational ocean analysis and forecasting systems, MO and SCSOFS, have been built based on the state-of-the-art hydrodynamic ocean model in France and China, respectively. This paper demonstrated the results of comparison and validation for the performance of both systems on the ocean circulation, the structures of the TS, and mesoscale activities in the SCS, based on the observed satellite and *in-situ* data in 2012. The comprehensive performances for the both systems are summarized as follow.

Both systems have capabilities to simulate the main basin-scale circulations in the SCS and model results are in good agreement with the  AVISO data. And MO has better performance than SCSOFS in simulating several main branches of the SCS ocean circulations and eddies in January. SCSOFS did not generate many mesoscale or small scale circulations , which may be cause by a little strong horizontal mixing set in the model. The westward intensification in the eastern coast of the Vietnam is most strong in autumn among the four seasons. For the type of the Kuroshio intruding the SCS, the AVISO observations, and model  results from both MO and SCSOFS show the looping path in winter, the leaping path in summer and leaking path in autumn. However, the leaking path, looping path and leaping path are shown for AVISO, MO and SCSOFS in spring.

Both systems demonstrated  the similar variation trends in u and v components time series with the mooring observation. The RMSE between MO, SCSOFS and mooring observation are 0.075

m/s, 0.094 m/s for u-component, 0.062 m/s, 0.084 m/s for v-component, respectively. The results of MO

are in better agreement with the observation than those of SCSOFS, especially during the period of the

Typhoon Kai-tak.

The maximum, minimum, and mean for the basin-averaged 12 monthly RMSEs between MO and

MGDSST are 0.78 ℃, 0.37 ℃, 0.51 ℃, between SCSOFS and MGDSST are 1.15 ℃, 0.56 ℃, 0.86 ℃

in the SCS, respectively. For the horizontal and vertical distributions of TS, both systems have achieved  the same structures with the *in-situ* data, but the results of SCSOFS are in better agreement with the *in-situ* observations than those of MO. The correlation coefficients are 0.987 and 0.982 for temperature,  0.717 and 0.897 for salinity, between model results from MO and SCSOFS and *in-situ* data, over the 95% significance level, respectively. It indicates that the good relativity between MO, SCSOFS and the *in-situ* observations, the relativity of temperature is better agreement with *in-situ* data than those of salinity for both model results from MO and SCSOFS, and the result from SCSOFS is better agreement with *in-situ* data than MO for salinity.

The similar SST frontal patterns with their evident seasonal variations are shown in both systems. Most fronts achieve maximum strength in winter, become weak obviously due to the weakening of northeaster monsoon EAM in spring and summer, fade and disappear in autumn, which is consistent with the result of Wang et al. (2001).

During the typhoon Tembin in the NSCS, the obviously decreasing of SST about 2-4 ℃ occurs 1-day after typhoon passing shown in the results of MO, SCSOFS and OISST, which is consistent with previous studies. SCSOFS is in much better agreement with OISST than MO  for both  range and domain of SST decreasing.

MO has more eddies formed near the middle of SCS than AVISO, especially for anti-cyclones. SCSOFS

has more anti-cyclones , but less cyclones than AVISO. AVISO data and model results from

NO and SCSOFS all  show most eddies in spring and lest in fall, and less cyclones than anti-cyclones in summer. Both AVISO and SCSOFS have more cyclones than anti-cyclones in spring and fall.

In order to improve  performance of MO and SCSOFS in the SCS in future based on the results of,  the comparison and validation for the two systems, some recommendations are proposed as below:

For MO, we would like to suggest (1) to modify the model bathymetry in the coast area for the depth less than 200m to improve the model performance in shallow water area, such as SST front; (2) to change the initial conditions of TS to improve the TS vertical structures, especially for the salinity in deep water area;

For SCSOFS, we would like to suggest (1) to weaken horizontal mixing to get more reasonable mesoscale or small scale circulations; (2) to optimize the data assimilation scheme further to better assimilate the *in-situ* and satellite data; (3) to replace the surface forcing data with the higher horizontal or temporal resolution; (4) to replace the boundary conditions from monthly to weekly or daily like

MO. For both systems, we also would like to suggest to try to get and assimilate more observed data during the typhoon period to catch the typhoon process more exactly.

**Author contribution**

X. Zhu, H. Wang and G. Liu compared and validated the model results on velocities and TS. C.

Régnier and M. Drévillon build the MO, D. Wang build the SCSOFS. X. Kuang analyzed the model results on mesoscale eddy. S. Ren analyzed the model results on SST front. Z. Jing provided the *in-situ*

data. X. Zhu prepared the manuscript with contributions from all co-authors.

**Acknowledgements**

We would like to thank the anonymous reviewers and the Editor, Ivan Federico, for their valuable contributions that allow us to improve the manuscript substantially. This study is supported by the

National Natural Science Foundation of China under contract No. 41222038, 41376016, 41206023 and the Strategic Priority Research Program of the Chinese Academy of Sciences Grant No. XDA11020104

03.

[revised manuscript text omitted]